# `Flex-MoE`: Modeling Arbitrary Modality Combination via the Flexible Mixture-of-Experts

**Sukwon Yun**[1], **Inyoung Choi**[2], **Jie Peng**[3], **Yangfan Wu**[3], **Jingxuan Bao**[2],
**Qiyiwen Zhang**[2], **Jiayi Xin**[2], **Qi Long**[2], **Tianlong Chen**[1]

[1]University of North Carolina at Chapel Hill
[2]University of Pennsylvania
[3]University of Science and Technology of China

{swyun, tianlong}@cs.unc.edu, {inyoungc, jiayixin}@seas.upenn.edu,
{pengjieb, ustc_wyf}@mail.ustc.edu.cn
{jingxuan.bao, qiyiwen.zhang}@pennmedicine.upenn.edu, qlong@upenn.edu

## Abstract

Multimodal learning has gained increasing importance across various fields, offering the ability to integrate data from diverse sources such as images, text, and personalized records, which are frequently observed in medical domains. However, in scenarios where some modalities are missing, many existing frameworks struggle to accommodate arbitrary modality combinations, often relying heavily on a single modality or complete data. This oversight of potential modality combinations limits their applicability in real-world situations. To address this challenge, we propose `Flex-MoE` (Flexible Mixture-of-Experts), a new framework designed to flexibly incorporate arbitrary modality combinations while maintaining robustness to missing data. The core idea of `Flex-MoE` is to first address missing modalities using a new missing modality bank that integrates observed modality combinations with the corresponding missing ones. This is followed by a uniquely designed Sparse MoE framework. Specifically, `Flex-MoE` first trains experts using samples with all modalities to inject generalized knowledge through the generalized router ($\mathcal{G}$-Router). The $\mathcal{S}$-Router then specializes in handling fewer modality combinations by assigning the top-1 gate to the expert corresponding to the observed modality combination. We evaluate `Flex-MoE` on the ADNI dataset, which encompasses four modalities in the Alzheimer's Disease domain, as well as on the MIMIC-IV dataset. The results demonstrate the effectiveness of `Flex-MoE`, highlighting its ability to model arbitrary modality combinations in diverse missing modality scenarios. Code is available at: https://github.com/UNITES-Lab/flex-moe.

## 1 Introduction

In many fields, including healthcare, language, and vision, multimodal learning [6, 75, 37, 44] has emerged as a crucial approach for integrating data from multiple sources such as clinical records, imaging, and genetic data. Multimodal data enables more comprehensive analysis and decision-making, offering the potential for improved diagnosis and prediction in various applications [59, 33, 68]. However, a prominent challenge across these domains is the missing modality scenario [76, 60], where not all modalities are consistently available for every instance due to diverse reasons such as individualized data collection protocols or the variable availability of certain modalities.

38th Conference on Neural Information Processing Systems (NeurIPS 2024).

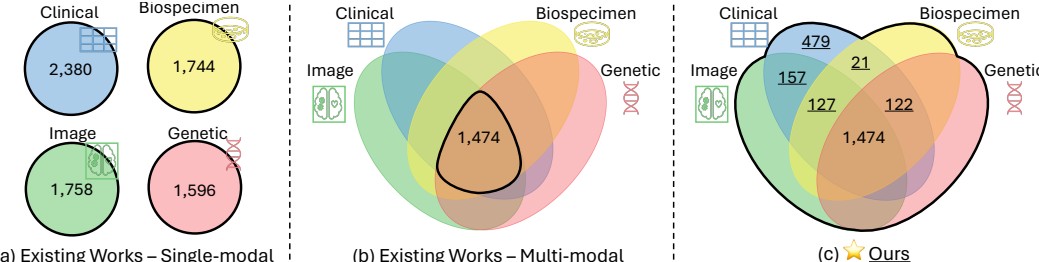

**(a) Existing Works – Single-modal** | **(b) Existing Works – Multi-modal** | **(c) ★ Ours**

Figure 2: Data statistics from a real-world multimodal dataset (e.g., the Alzheimer's Disease Neuroimaging Initiative (ADNI)), where patients exhibit unique combinations of available modalities. Existing approaches focus on either (a) single-modality data or (b) complete multimodal data, losing the potential to leverage other combinations. Our approach incorporates all possible modality combinations, offering a more robust solution to the missing modality scenario.

As an representative example, in Alzheimer's Disease (AD) [4], one of the most prevalent neurodegenerative disorders, handling this inherently multimodal data is crucial for accurate diagnosis (Figure 1). AD datasets often include a combination of clinical symptoms, imaging data [42], and genetic profiles [48]. However, in real-world clinical settings, not all these modalities are readily available for each patient. Some data, such as clinical and imaging data, may be available from routine visits, whereas other data, such as genetic or biospecimen information, may require additional time to collect. This leads to incomplete datasets, which poses a challenge for existing models that tend to either rely heavily on single modalities or only utilize complete data, thereby missing the opportunity to leverage the full potential of multimodal learning (Figure 2).

Figure 1: Multimodal AD.

**Single Modality and Complete Data Reliance.** The reliance on single-modality data or complete data across many frameworks is a significant limitation in real-world scenarios, where missing data is the norm rather than the exception. As seen in Figure 2, many current models either work with single-modality data or focus on the intersection of modalities, neglecting the potential contribution of partially available modalities. In healthcare, particularly for diseases such as AD, this often leads to missed opportunities in diagnosis and treatment due to the inability to fully exploit multimodal data when some modalities are missing.

**Oversight of Modality Combinations.** Beyond the challenge of missing modalities, there is also the need to model the interactions between available modalities properly. Different combinations of modalities can provide complementary information, and each combination may hold unique significance for downstream tasks. For example, in AD diagnosis, combining biospecimen data and imaging data can reveal key insights: cerebrospinal fluid biomarkers may indicate early signs of AD [21], while functional MRI can highlight cognitive impairments [42]. Hence, it is essential to develop models that not only handle missing modalities but also effectively utilize the available modality combinations.

Given the general challenge of the missing modality scenario in multimodal learning, we propose a novel framework, `Flex-MoE` (Flexible Mixture-of-Experts), to flexibly incorporate arbitrary modality combinations while maintaining robustness to missing data. `Flex-MoE` first sort samples based on the available modalities and process them through modality-specific encoders. For missing modalities, we introduce a learnable missing modality bank, which provides learnable embeddings for missing modalities based on the observed ones. This approach ensures that the model can handle incomplete datasets effectively. Our framework also builds upon Sparse Mixture-of-Experts (SMoE) design, allowing us to generalize the expert knowledge from complete data (samples with all modalities) through the $\mathcal{G}$-Router, followed by a specialized $\mathcal{S}$-Router for handling fewer modality combinations. Each expert becomes specialized in handling different modality combinations, ensuring that the model can effectively process any combination of modalities. We demonstrate the effectiveness of `Flex-MoE` through comprehensive experiments on several real-world datasets, including the Alzheimer's Disease Neuroimaging Initiative (ADNI), which involves four key modalities for AD stage prediction, and the MIMIC-IV dataset. The results confirm the robustness of `Flex-MoE` in diverse missing modality scenarios.

The contributions of this work can be summarized as follows:

⋆ We introduce a flexible framework that effectively incorporates arbitrary modality combinations and addresses the missing modality scenario across various domains.

⋆ `Flex-MoE` features a novel approach, including a missing modality bank and generalized and specialized expert training, which ensures robustness to missing modality scenario.

⋆ Extensive experiments on real-world datasets, including ADNI and MIMIC-IV, showcase the consistent and robust performance of `Flex-MoE` in handling diverse modality combinations.

## 2 Related Works

**Single Modality Approach** In many fields, deep learning models often rely on single modality data for tasks such as classification [15, 13, 31, 71], diagnosis [56, 72], or prediction [12, 61, 34]. While effective in certain cases, these approaches fail to capture the potential synergies between different data sources, especially in contexts where multiple modalities are available. In the Alzheimer's Disease domain, many studies focus on specific modalities. For instance, image-based approaches include a VGG19 model [43] that diagnoses early-stage AD from MRI scans and a modified ResNet18 architecture [45] that predicts AD progression using fMRI data. Other studies focus on genomics, such as DLG [36] for classifying AD patients and SWAT-CNN [26] for discovering AD-associated genetic variants. In the biospecimen modality, a deep learning-assisted spectroscopy platform [29] diagnoses AD by analyzing blood-based amyloid-beta and metabolite biomarkers. Regarding clinical data, a deep learning model [5] outperforms earlier machine learning techniques in classifying AD patients. However, since AD data is inherently multimodal, methods based on a single modality are suboptimal, missing the potential to leverage interactions between different modalities.

**Multimodal Approach** Across multiple fields, multimodal learning has become increasingly valuable for its ability to integrate and capture dynamics within and across different modalities, providing richer and more comprehensive representations of data. Approaches such as the Tensor Fusion Network [74], Multimodal Transformer [58], and Multimodal Adaptation Gate [53] highlight the effectiveness of combining multiple data sources. Recently, sparse mixture-of-experts-based methods, such as [44, 8, 19], have been introduced to enhance modality interactions, though these methods are still relatively unexplored in the AD domain due to the complexity of handling various modality combinations. In AD research, some works have emerged to leverage multimodal data, such as [46] and [59], which integrated a deep learning framework that combines imaging, genetic, and clinical data, achieving superior AD staging accuracy. Another multimodal model [33], incorporating longitudinal and cross-sectional data, provided more accurate AD predictions. While multimodal AD studies have shown significant progress, the challenge of missing modalities, especially in the context of how to effectively cope with modality combinations, remains largely underexplored.

## 3 Methods

### 3.1 Preliminaries and Notations

**Why Sparse Mixture-of-Experts?** Given a multimodal nature, we choose to utilize Sparse Mixture-of-Experts (SMoE) [55] due to its computational efficiency and its ability to handle multimodal data by effectively alleviating the gradient conflict optimization issue between modalities [50]. To briefly introduce SMoE, Traditional Mixture-of-Experts (MoE) models [23, 28, 9, 70] evolved by incorporating sparsity into their structure, optimizing computational efficiency and model performance. SMoE selectively activates only the most relevant experts for a given task, reducing overhead and improving scalability. This innovation is particularly beneficial in handling complex, high-dimensional datasets across diverse applications. It has been widely used in vision [54, 40, 16, 2, 18, 62, 69, 1, 49] and language processing [35, 30, 78, 77, 79, 25] fields, dynamically assigning different parts of the network to specific tasks [41, 3, 20, 11] or data modalities [32, 44]. Research has shown its effectiveness in classification tasks in digital number recognition [20] and medical signal processing [3]. In this work, we further explore the use of SMoE to model arbitrary modality combinations and address the missing modality scenario.

**Notation.** Formally, the SMoE consists of multiple experts, denoted as $f_1, f_2, \ldots, f_{|E|}$, where $|E|$ represents the total number of experts, and a router, $\mathcal{R}$, which is responsible for the routing mechanism and sparsely selects the top-$k$ experts. For a given embedding or token $\mathbf{x}$, the router $\mathcal{R}$

engages the top-$k$ experts based on the highest scores obtained from softmax function with learnable gating function, $g(\cdot)$ (usually one or two layer MLP), and output $\mathcal{R}(\mathbf{x})_i$, where $i$ denotes the expert index. This process can be described as follows:

$$
\mathbf{y} = \sum_{i=1}^{|E|} \mathcal{R}(\mathbf{x})_i \cdot f_i(\mathbf{x}),
$$
$$
\mathcal{R}(\mathbf{x}) = \text{Top-K}(\text{softmax}(g(\mathbf{x})), k), \tag{1}
$$
$$
\text{TopK}(\mathbf{v}, k) = \begin{cases} \mathbf{v}, & \text{if } \mathbf{v} \text{ is in the top } k, \\ 0, & \text{otherwise.} \end{cases}
$$

## 3.2 Our approach: `Flex-MoE`

In this section, we present our novel algorithm, `Flex-MoE`, specifically designed to flexibly address the challenge of missing modalities in the multimodal domain. We start by sorting the samples based on their number of observed modalities. Following a modality-specific encoder, we supplement the embeddings for missing parts via missing modality bank completion. This effectively manages missing modalities by learning embedding banks that capture the information specific to observed modality combinations. Next, a Transformer coupled with an SMoE layer is employed. We introduce an expert generalization and specialization step to optimize modality utilization by fully leveraging samples with complete modalities and obtaining modality combination-specific knowledge through samples with fewer modalities. A comprehensive illustration of `Flex-MoE` is provided in Figure 3. Throughout the details in the following section, while our work is exemplified through the AD domain for predicting AD stages using four representative modalities—image, clinical, biospecimen, and genetic—it is important to note that `Flex-MoE` can be generalized to any other multimodal domain.

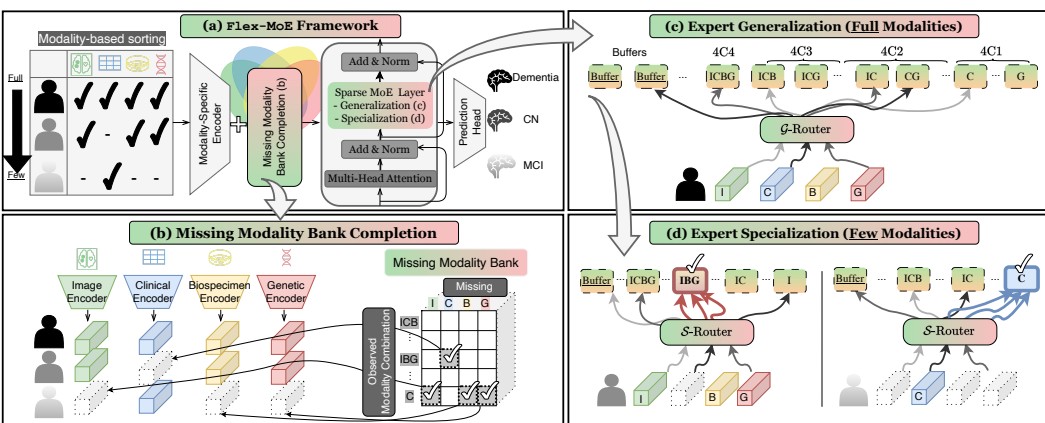

Figure 3: The comprehensive illustration of our proposed methodology, `Flex-MoE`. (a) Overall framework of `Flex-MoE`. Given samples with diverse modality combinations, we first sort the samples based on their number of available modalities in descending order, and then pass through the modality-specific encoder. (b) Each encoder is only trained with their available samples. For the missing embeddings, we introduce a missing modality bank containing learnable embeddings given the observed modality combination with their corresponding missing modality index. Equipped with this embedding, `Flex-MoE` passes through the Transformer where the FFN layer is replaced with a Sparse MoE layer. Here, (c) full modality samples take charge of training generalized experts in a balanced manner via $\mathcal{G}$-router, then (d) the remaining few modality combinations further specialize the expert knowledge with $\mathcal{S}$-Router, which fixes the top-1 gate as the corresponding observed modality combination expert. In this figure, top-2 selection of experts is illustrated as an example.

### 3.2.1 Missing Modality Bank Completion

Given a set of samples with their own modalities, it is straightforward to pass them through modality-specific encoders, such as a 3D-CNN for MRI images. However, we are dealing with a *missing*

*modality scenario* in multimodal data, where specific modalities are often missing based on their observed modality combinations. Thus, it is common to use padded or imputed inputs for the corresponding missing modalities in a multimodal setting. This approach becomes troublesome when considering interactions between modalities. For instance, given a batch with samples, some samples might have image, genetic, and clinical modalities, while others might be missing image and genetic modalities. In such cases, the image encoder and genetic encoder would take zero-padded or hypothesized imputed inputs derived from the missing samples, which are synthetic and of lower quality than the observed ones. This negatively affects the training of modality-specific encoders. Additionally, heavy imputation for each modality in a multimodal setting increases time complexity, which is not desirable in clinical settings.

Given this situation, we propose training each encoder *solely with observed samples* to fully leverage the potential of the encoder by using only observed inputs. Unlike existing approaches, our design principle considers modality combinations to ensure robust and flexible training and effective handling of missing modalities. As illustrated in Figure 1, each patient has diverse symptoms and personalized treatments, leading to variations in available modalities. For example, patients might lack image and genomic modalities (i.e., possessing only biospecimen and clinical modalities) due to various reasons such as patient conditions, resource limitations, or specific clinical settings [22, 51]. Imputing these missing modalities must be handled within this context, rather than applying a global learnable representative embedding for each modality without considering the observed environment.

Therefore, we propose a learnable missing modality bank. Given the number of modality combinations without fully observed scenarios, the total cases would be, given a modality set $\mathcal{M} = \{\mathcal{I}, \mathcal{C}, \mathcal{B}, \mathcal{G}\}$, $\sum_{m=1}^{|\mathcal{M}|-1} \binom{|\mathcal{M}|}{m} = 2^{|\mathcal{M}|} - 1$. The resulting missing modality bank can be defined as $\mathbf{B} \in \mathbb{R}^{2^{|\mathcal{M}|}-1 \times |\mathcal{M}|}$. By using this bank, the concatenated embedding of all modalities for patient $i$, $\mathbf{h}_i = [\mathbf{e}_i^{\mathcal{I}}, \mathbf{e}_i^{\mathcal{C}}, \mathbf{e}_i^{\mathcal{B}}, \mathbf{e}_i^{\mathcal{G}}]$ would be represented as follows:

$$\mathbf{e}_i^m = \begin{cases} \text{Encoder}^m(i) & \text{if modality } m \text{ is observed in sample } i \\ \mathbf{B}_{\mathcal{M}\backslash m, m} & \text{otherwise} \end{cases}, \quad \forall m \in \{\mathcal{I}, \mathcal{C}, \mathcal{B}, \mathcal{G}\} \quad (2)$$

where $\mathbf{e}_i^m \in \mathbb{R}^d$ denotes the embedding from the corresponding modality $m$ for patient $i$, and $d$ is the hidden dimension. Here, the main idea of the missing modality bank completion is to supplement missing modalities from a predefined bank, ensuring robust data integration with observed ones. For example, if a patient lacks clinical data but has imaging, biospecimen, and genetic data, the observed modalities pass through their specific encoders. The missing clinical embedding is supplemented from the missing modality bank, indexed by the observed modalities (e.g., $\mathbf{B}_{\mathcal{M}\backslash m, m} = \mathbf{B}_{\{\mathcal{I}, \mathcal{G}, \mathcal{B}\}, \mathcal{C}}$). By doing so, the encoder for each modality can be trained without encountering non-observed, incomplete input features. Then, we move on to the Transformer layer, where the FFN layer is replaced by an SMoE layer, a common approach in the SMoE domain [24, 38, 10].

### 3.2.2 Expert Generalization & Specialization

While adopting the SMoE backbone, it is important to note that our environment differs from concurrent SMoE studies, especially in terms of multimodal learning with missing modalities. In this context, choosing the most relevant tokens is challenging for experts, since the significance, i.e., the quality of input information, varies with the missing modalities. This significantly motivates us to take a distinct approach from concurrent SMoE studies, where the input token is derived from fully observed scenarios. To address the unique challenges of the missing modality scenario, we propose a modality combination-specific MoE design. Specifically, we assign expert indices based on all possible modality combinations. For example, 'IGCB' is assigned as 0, 'IGC' as 1, ..., up to 'B' as 14. The remaining experts act as buffers, allowing the Router to select the most relevant top-$k$ experts and activate them automatically. This approach leaves room for flexibility and maintains the initial intuition of the MoE design.

**Generalization** It now becomes clear why the samples used for training `Flex-MoE` are sorted in descending order. Inspired by curriculum learning [7, 63], where easy samples are presented first and more challenging samples appear later, we regard the level of difficulty as the number of missing modalities. We first train our SMoE layer with easy samples, where all modalities are fully observed. Using this intersection as a gold standard, we initially train all the experts in the MoE model. The procedure essentially follows the vanilla SMoE design as described in Equation 1, but with one key

difference: the input tokens consist only of inputs where all modalities are fully observed. Hence, we refer to this router as the Generalized Router, $\mathcal{G}$-Router. This approach leverages the completeness of information in these samples, which should be fully utilized before specializing the experts in their respective areas. To ensure balanced activation of the experts initially, which will later specialize, we incorporate the load and importance balancing loss [55], which will later be exemplified in Equation 4.

**Specialization** Once the experts are initially trained using fully observed samples, we aim to specialize each expert, which is the key advantage of the MoE design. We leverage the remaining samples, which encompass diverse modality combination configurations. Each modality combination requires its own specialized expertise. For instance, samples with Image, Biospecimen, and Genetic data will have a corresponding expert index activated through the top-1 gating mechanism to fully utilize the specialized knowledge of that expert (i.e., expert 'IBG' in Figure 3). To effectively specialize the modality combination-specific experts, we propose a Specialized Router design, $\mathcal{S}$-Router, which encompasses following technical novelties. First, to facilitate targeted expert selection when an input token is provided, we avoid manually replacing the selected routing policy with our preferred choice in a post-hoc manner, which would stop the continuous gradient flow. Instead, we innovatively introduce a cross-entropy loss between the top-1 expert selection and the targeted expert indices for each token by the $\mathcal{S}$-Router. Formally, this can be described as follows:

$$\mathcal{L}_{ce} = -\sum_{j=1}^{n} \mathcal{MC}(\mathbf{x}_j) \log(\max(\mathcal{S}\text{-Router}(\mathbf{x}_j))) \tag{3}$$

where $\mathcal{MC}(\mathbf{x}_j)$ denotes the modality combination index of a given token $\mathbf{x}_j$ in a total of $n$ tokens. $\max(\mathcal{S}\text{-Router}(\mathbf{x}_j))$ denotes the maximum prediction probability of the corresponding activated expert index, which corresponds to the probability of the top-1 expert index. By comparing these two, the router is trained to activate the corresponding expert index for a given input token with a certain modality combination. Thus, the specialized knowledge inherent in specific modality combinations is naturally contained within the target expert.

Moreover, when calculating load and importance balancing loss [55], we specifically compute the loss for the *remaining* top-$k$-1 experts, as the top-1 selection is manually handled and thus considered biased rather than balanced. We aim for the selection of the remaining $k$-1 experts to occur in a balanced manner, allowing interaction with other related modality combinations. Formarlly, it can be expressed as follows:

$$\mathcal{L}_{\text{balance}} = \text{CV}^2 \left( \sum_{j}^{N} \text{importance}_j \right) + \text{CV}^2 \left( \sum_{j}^{N} \text{load}_j \right)$$

$$\text{where } \text{importance}_e = \sum_{i}^{N} g_{ie}, \quad \text{load}_e = \sum_{i}^{N} \delta(g_{ie} > 0), \quad \forall e \in E \setminus e_{\text{top-1}} \tag{4}$$

where $\text{CV}^2(x) = \left( \frac{\sigma(x)}{\mu(x)} \right)^2$, $\sigma(x)$ is the standard deviation of $x$, $\mu(x)$ is the mean of $x$, $g_{ie}$ is the gate value for sample $i$ with expert index $e$ as discussed in Equation 1, and $\delta(\cdot > 0)$ is an indicator function that is 1 when the inner value is greater than 0. $E \setminus e_{\text{top-1}}$ denotes the set of expert indices excluding the top-1 selected expert index. This ensures that the resulting MoE model not only retains global knowledge but also incorporates specialized expert knowledge tailored to specific modality combinations. During the inference phase, the specified expert index for a particular modality combination can be activated alongside other experts, enabling predictions to consider both the specific modality combination and intersections with other modalities.

Finally, the output embeddings of the Sparse MoE layer pass through a 1-layer MLP prediction head to predict one of the three stages of AD, i.e., Dementia, CN, or MCI. To further facilitate a curriculum-learning approach, we first use warm-up epochs with sorted samples, followed by shuffled samples for the remaining epochs. This strategy enhances generalizability across diverse input samples, enabling better handling of variability during the inference phase.

# 4 Experiments

## 4.1 Experimental Setting

**ADNI Dataset** Alzheimer's Disease Neuroimaging Initiative (ADNI) is a landmark multimodal AD dataset that tracks disease progression and pathological changes, comprising of comprehensive imaging, genetic, clinical, and biospecimen data ([64], [67]). The imaging data in ADNI includes magnetic resonance imaging (MRI) and positron emission tomogrpahy (PET). The genetic data includes a variety of genetic information, including genotyping data such as APOE genotyping and single nucleotide polymorphisms. The clinical data includes demographics, physical examinations, and cognitive assessments. Biospecimens such as blood, urine, and cerebrospinal fluid are also collected. ADNI has established standardized multi-center protocols and provides open access to qualified researchers, making it a gold-standard resource in the field ([65], [66]). Before integrating all modalities, to address the initial missing data within each modality, we applied simple mean imputation [39] for each column. For more detailed data table with preprocessing steps for each modality, please refer to Appendix A.1.

**MIMIC-IV Dataset** We use the Medical Information Mart for Intensive Care IV (MIMIC-IV) database [27], which contains de-identified health data for patients who were admitted to either the emergency department or stayed in critical care units of the Beth Israel Deaconess Medical Center in Boston, Massachusetts24. MIMIC-IV excludes patients under 18 years of age. We take a subset of the MIMIC-IV data, where each patient has at least more than 1 visit in the dataset as this subset corresponds to patients who likely have more serious health conditions. For each datapoint, we extract ICD-9 codes, clinical text, and labs and vital values. Using this data, we perform binary classification on one-year mortality, which foresees whether or not this patient will pass away in a year. We drop visits that occur at the same time as the patient's death. In order to align the experimental setup with the ADNI data, which does not contain temporal data, we take the last visit for each patient.

**Baselines** We compare the performance of `Flex-MoE` with various state-of-the-art baselines across modality-specific, e.g., image or genetic, and multimodal approaches. For the image-only modality, we first experimented with 3D MRI scans by utilizing a 3D CNN [17] and an architecture that combines 3D CNN and 3D CLSTM [68]. To decrease computational complexity, we also extracted 2D slices from the 3D volumes. For 2D MRI scans, we implemented the VGG architecture with pre-trained weights and applied layer-wise transfer learning [43], as well as a modified ResNet-18 network [45]. For the genetic-only approach, we employed a ResNet-34 based architecture to handle the high-dimensional genetic data [36]. In ADNI dataset, we further implemented domain speicfic baselines, such as auto-encoder and 3D CNN-based architecture that incorporates imaging, genetic, and clinical data [59], and a GRU-based architecture that considers imaging, genetic, clinical, and biospecimen data [33]. Moreover, we include ShaeSpec [60], which utilizes a spectral attention mechanism to emphasize important features across modalities, and mmFormer [76], which is based on transformer-based multimodal fusion with an attention mechanism. For multimodal approaches in both ADNI and MIMIC-IV, we incorporate the recent FuseMOE [19] model, which directly integrates multimodal data through a mixture of experts strategy, as the most straightforward baseline. Additionally, we compare the following methods: MulT [57], which captures cross-modal interactions through cross-attention mechanisms; MAG [52], which fuses multimodal features by mapping them to an adaptation vector; TF [73], which combines multimodal embedding sub-networks and a tensor fusion layer; and LIMoE [44], which addresses training stability in multimodal learning using entropy regularization based on contrastive learning.

**Experimental Settings.** To ensure a fair comparison with other baselines, we used the best hyper-parameter settings provided in the original papers. If not available, we tuned the learning rate in 1e-3, 1e-4, 1e-5, the hidden dimension in 64, 128, 256, and the batch size in 8, 16. For our proposed method, we searched the number of experts in 16, 32, and Top-$k$ in 2, 3, 4. We set the coefficient of the sum of additional losses (importance and load balancing) combined with our cross-entropy loss to 0.01, scaling it within the task classification loss. For the dataset split, we chose 70% for training, with the remaining 30% split evenly between validation and test sets (15% each). It is important to note that, to share the same inference space, where single and multimodal baselines should both be able to predict, we opted to choose the intersection as the test and validation sets. This means that during the training phase, the dataset can be incomplete. For the multi-modal baselines, if they had the ability to impute or interact with other modalities, we leveraged their methods. Otherwise, we used zero-padding to facilitate batch-wise training. For single-modal and multi-modal baselines

that solely work on the intersection region, we filtered that data and used it during training. All experiments were conducted using NVIDIA A100 GPUs. Each experiment was run three times with different seeds to ensure reproducibility, and the results were averaged. The optimal hyperparameter settings for `Flex-MoE` can be found in Appendix A.2.

Table 1: Performance on ADNI dataset with ACC metric across different models and modality combinations, given the Image ($\mathcal{I}$, 🧠), Genetic ($\mathcal{G}$, 🧬), Clinical ($\mathcal{C}$, ▦), and Biospecimen ($\mathcal{B}$, 🔬) modalities. $\mathcal{MC}$ denotes observed modality combination.

| $\mathcal{MC}$ | 🧠 | 🧬 | ▦ | 🔬 | [59] | [33] | ShaSpec | mmFormer | TF | MulT | MAG | LIMoE | FuseMoE | Flex-MoE |
|---|---|---|---|---|---|---|---|---|---|---|---|---|---|---|
| | | | | | | | | **Dataset: ADNI / Metric: ACC** | | | | | | |
| $\mathcal{I},\mathcal{G}$ | • | • | | | 54.81 ±1.45 | 53.59 ±2.98 | 48.09 ±0.66 | 49.85 ±4.92 | 59.94 ±0.40 | 60.32 ±0.95 | 59.94 ±1.00 | 59.29 ±0.95 | 60.41 ±0.87 | **61.08** ±0.78 |
| $\mathcal{I},\mathcal{C}$ | • | | • | | 44.35 ±1.99 | 57.15 ±1.58 | 47.62 ±1.81 | 51.96 ±4.23 | 54.53 ±0.66 | 50.14 ±1.05 | 52.19 ±2.90 | 52.38 ±3.46 | 53.13 ±1.97 | 56.49 ±2.55 |
| $\mathcal{I},\mathcal{B}$ | • | | | • | 40.80 ±2.94 | 57.61 ±1.86 | 50.98 ±2.09 | 51.45 ±3.53 | 52.57 ±2.06 | 51.17 ±2.88 | 52.47 ±4.11 | 53.87 ±2.75 | 49.67 ±1.97 | **60.41** ±0.26 |
| $\mathcal{G},\mathcal{C}$ | | • | • | | 51.91 ±1.39 | 52.85 ±2.47 | 52.85 ±2.65 | 49.58 ±4.45 | 38.38 ±3.03 | 46.03 ±5.42 | 40.34 ±6.11 | 35.76 ±6.24 | 38.84 ±2.42 | **60.60** ±0.26 |
| $\mathcal{G},\mathcal{B}$ | | • | | • | 45.01 ±1.30 | 52.66 ±3.63 | 58.54 ±2.97 | 48.45 ±4.56 | 42.20 ±1.78 | 39.40 ±2.91 | 40.52 ±2.52 | 36.88 ±5.04 | 37.91 ±0.80 | **63.59** ±1.04 |
| $\mathcal{C},\mathcal{B}$ | | | • | • | 44.63 ±0.92 | 63.68 ±0.48 | 59.10 ±2.69 | 47.71 ±4.49 | 39.68 ±2.38 | 44.54 ±0.82 | 40.15 ±2.58 | 43.98 ±0.00 | 37.91 ±0.80 | 60.50 ±0.82 |
| $\mathcal{I},\mathcal{G},\mathcal{C}$ | • | • | • | | 55.12 ±2.38 | 54.72 ±0.28 | 49.30 ±3.17 | 46.49 ±3.57 | 54.06 ±1.98 | 60.97 ±0.95 | 61.34 ±0.61 | 53.50 ±2.25 | 60.97 ±1.32 | **63.21** ±1.73 |
| $\mathcal{I},\mathcal{G},\mathcal{B}$ | • | • | | • | 56.12 ±3.44 | 55.28 ±3.44 | 52.85 ±0.53 | 47.15 ±6.43 | 54.44 ±2.26 | 53.03 ±1.95 | 54.15 ±1.06 | 53.97 ±1.08 | 52.85 ±1.00 | **62.28** ±2.75 |
| $\mathcal{I},\mathcal{C},\mathcal{B}$ | • | | • | • | 43.79 ±0.69 | 60.97 ±2.60 | 52.85 ±3.30 | 47.18 ±4.68 | 52.29 ±1.47 | 49.86 ±1.50 | 53.24 ±0.50 | 54.97 ±0.00 | 49.67 ±1.00 | **64.05** ±1.78 |
| $\mathcal{G},\mathcal{C},\mathcal{B}$ | | • | • | • | 45.28 ±1.85 | 53.87 ±3.35 | 62.09 ±3.27 | 46.38 ±4.24 | 43.33 ±4.43 | 43.32 ±6.74 | 37.25 ±1.99 | 40.99 ±2.62 | 34.64 ±1.95 | **65.36** ±1.38 |
| $\mathcal{I},\mathcal{G},\mathcal{C},\mathcal{B}$ | • | • | • | • | 52.10 ±0.99 | 55.64 ±1.86 | 52.84 ±0.53 | 58.92 ±6.58 | 57.24 ±3.05 | 58.82 ±0.82 | 61.44 ±1.61 | 55.18 ±4.22 | 59.52 ±1.00 | **66.11** ±1.14 |

Table 2: Performance on MIMIC-IV dataset with ACC metric across different models and modality combinations, given the Lab and Vital values ($\mathcal{L}$, 🗡), Clinical Notes ($\mathcal{N}$, 📄), and ICD-9 Codes ($\mathcal{C}$, 📋) modalities. $\mathcal{MC}$ denotes observed modality combination.

| $\mathcal{MC}$ | 🗡 | 📄 | 📋 | TF | MulT | MAG | LIMoE | FuseMoE | Flex-MoE |
|---|---|---|---|---|---|---|---|---|---|
| | | | | | **Dataset: MIMIC-IV / Metric: ACC** | | | | |
| $\mathcal{L},\mathcal{N}$ | • | • | | 60.05 ±1.96 | 57.96 ±7.25 | 62.72 ±2.36 | 63.80 ±1.99 | 60.50 ±3.82 | **76.14** ±0.73 |
| $\mathcal{L},\mathcal{C}$ | • | | • | 64.13 ±3.39 | 62.47 ±2.01 | 60.13 ±1.97 | 64.89 ±1.46 | 63.31 ±3.21 | **75.15** ±0.55 |
| $\mathcal{N},\mathcal{C}$ | | • | • | 60.97 ±2.36 | 62.23 ±2.81 | 59.41 ±4.15 | 64.27 ±4.05 | 64.77 ±3.05 | **74.96** ±1.59 |
| $\mathcal{L},\mathcal{N},\mathcal{C}$ | • | • | • | 63.11 ±2.17 | 64.62 ±0.44 | 62.87 ±2.50 | 61.61 ±2.37 | 63.90 ±1.72 | **76.81** ±0.90 |

## 4.2 Primary Results

In Table 1 and Table 2, we provide a comprehensive comparison of `Flex-MoE` with various multi-modal baselines. We have the following observations: **(1)** Overall, `Flex-MoE` performs effectively in diverse multimodal settings, fully harnessing its potential as more modalities become available. This is supported by the large margin of improvement (7.6% and 11.07% over the best performing baselines, MAG and the most recent work FuseMoE, respectively, in full modality settings in Table 1). **(2)** Although the recently proposed FuseMoE [19] suggested its ability to handle missing scenarios, the lack of effective modality combination creates a bottleneck in such AD domain, even performing worse when a smaller number of modalities is used (FuseMoE performs better with three modalities than with full modalities), which is not optimal given the diverse missing modality scenarios. **(3)** Despite its specific characteristics in the AD domain [33, 59], the reliance on intersection data and the lack of consideration for how missing modalities relate to observed modality combinations have been overlooked. **(4)** Overall, the performance gain derived from `Flex-MoE` can be attributed to its unique ability to cope with diverse modality combinations through a missing modality bank, and its capability to fully harness the knowledge of samples via a generalization followed by a specialization step for experts. For additional results on different metrics, such as Macro-F1 and AUC, please refer to Appendix A.3.

## 4.3 Effectiveness of Modality Combination Consideration

To validate the effectiveness of the two essential modules of `Flex-MoE`—the *missing modality bank* and the *unique SMoE design*—under a missing modality scenario, we evaluate them followingly.

First, to evaluate the effectiveness of the missing modality bank introduced in Figure 3 (b), we assess whether it captures relevant embedding information given an observed modality combination. Specifically, we validate this by examining the inter-relationship between modalities, focusing on

the consistency of how the missing modality bank handles missing information. In Figure 4, we measure the cosine similarity between observed modality combinations. The key observation is that **(1)** with more overlapping modality combination, it tends to share more similar embedding information. This is evident in the left side of Figure 4, where the full modality scenario (ICBG) shows higher similarity with ICB and CBG (0.56 and 0.58, respectively) compared to IC and CB (0.46 and 0.45). The clinical modality (C) is most similar across combinations, which aligns with the dataset characteristic that clinical data is present in all input combinations, as shown in Figure 2. On the right side of Figure 4, the similarity between missing modalities is shown. When **(2)** modality G is missing, it is more similar to the cases where C and B are missing compared to I, suggesting that certain missing modalities share more commonality in how they are handled by the model. This insight underscores the importance of careful consideration when modeling missing modalities and demonstrates how the missing modality bank effectively captures necessary embedding information in terms of modality combination.

Furthermore, in Figure 5, we show the activation ratio of input modality combinations across each expert index, i.e., possible modality combinations. We observe two key findings: **(1)** Thanks to expert generalization using full modality samples (BCGI), generalized knowledge is distributed across all experts. This allows each expert to leverage commonly shared knowledge while activating the most relevant inputs for the downstream task. **(2)** Expert specialization enables each expert to acquire specialized knowledge. For instance, in the case of the BCG expert, the two most activated input tokens were BCGI and its corresponding token, BCG. Similarly, for the BCI and CI experts, they not only possess general knowledge from BCGI but also hold their own specialized knowledge from BCI and CI, respectively, to effectively handle these inputs. This demonstrates that when a certain modality combination is provided, the top-1 expert is successfully selected, allowing it to supplement its specialized and necessary information. Overall, these routing experiments demonstrate that `Flex-MoE` contains both globally generalized and locally expert-specific knowledge, achieved by leveraging samples with both full and fewer modalities.

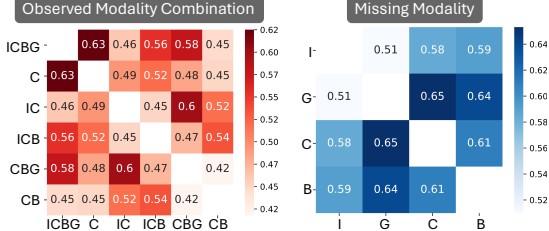

Figure 4: Cosine similarity between observed modality combination and missing modality, corresponding to row and column in missing modality bank.

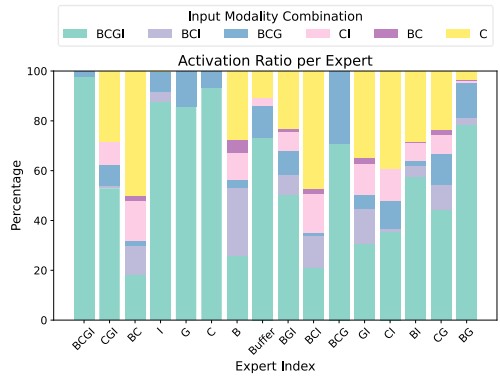

Figure 5: Modality combination activation ratio.

### 4.4 Comprehensive Evaluation

**Ablation Study.** In this section, we investigate the crucial components that contribute most positively to the performance gain of `Flex-MoE`. From Table 3, we observe that **(1)** when both expert specialization and generalization are absent, the performance drop is most severe. Additionally, **(2)** the performance decline in the embedding bank negatively affects overall performance, indicating that the missing modality bank combined with expert generalization and specialization is crucial for handling missing modality scenarios. Furthermore, (3) the sorting based on descending order appear most effective as the expert generalization occurs first withi the full modality samples.

**Sensitivity Study.** In Figure 6, we also varied the hyperparameters used in this study. We examined the number of experts, number of SMoE layers and top-$k$ selection. We found that **(1)** employing many experts does not always guraantee a higher performance compared to its increase in complexity, showing using 16 experts appear to be a suitable choice to equip fine-grained specialized knowledge. **(2)** Using sing a single layer of the SMoE was most effective, as stacking more layers or adding a Transformer block caused an overload to parameter learning. Additionally, **(3)** compared to the commonly used top-2 gating network in concurrent SMoE studies, we found that top-4 selection was

Table 3: Ablation study of `Flex-MoE`.

| | ACC | F1 |
|---|---|---|
| Flex-MoE | **66.11** | **64.73** |
| w/o ES | 62.75 | 60.79 |
| w/o {ES + EG} | 62.49 | 60.07 |
| w/o embedding bank | 63.87 | 62.48 |
| w/o sorting - random | 62.65 | 60.70 |
| w/o sorting - ascending | 63.87 | 62.22 |

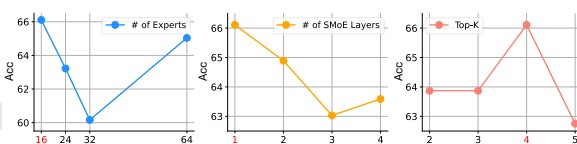

Figure 6: Sensitivity analysis of `Flex-MoE`. The hyper-parameters include the number of experts, the number of SMoE layers and Top-$k$ expert selection. For the experiment, ADNI dataset with full modalities is used.

the most effective. This is because manually assigning the top-1 expert index to the target modality combination leaves more room for better harmonization with the SMoE design.

**Complexity Study.** In Table 4, we further verify the benefits of utilizing the SMoE design in terms of mean time per iteration, GFLOPs, and the number of parameters compared to the baselines. We observed the following: **(1)** Compared to recent baseline FuseMoE, `Flex-MoE` achieves notable efficiency gains (e.g., 22.74%, 1.15%, and 89.17% gain in mean time, GFLOPs, and # of parameters, respectively.) while delivering higher performance. **(2)** Although TF appears to be a lightweight design in the $\mathcal{I}, \mathcal{G}$ and $\mathcal{I}, \mathcal{G}, \mathcal{C}$ settings, it trades off computational efficiency with significantly lower performance compared to `Flex-MoE`. **(3)** Notably, as the number of modalities increases, existing models tend to become more complex in terms of GFLOPs and the number of parameters to manage the additional complexity. However, `Flex-MoE` remains robust and efficient, maintaining higher performance due to its effective use of sparsely activated experts, brought by the SMoE framework.

| Modality | Metric | TF | MulT | MAG | LIMOE | FuseMoE | Flex-MoE |
|---|---|---|---|---|---|---|---|
| $\mathcal{I}, \mathcal{G}$ | Mean Time (s) (↓) | 12.40 | 12.85 | 11.64 | 12.65 | 18.68 | 12.73 |
| | GFLOPs (↓) | 59.05 | 59.24 | 59.06 | 59.24 | 59.74 | 59.06 |
| | # of Parameters (↓) | 33,370,898 | 37,343,683 | 36,454,595 | 37,344,707 | 264,680,387 | 36,516,807 |
| | Accuracy (↑) | 59.94 ±0.40 | 60.32 ±0.95 | 59.94 ±1.01 | 59.29 ±0.95 | 60.41 ±0.87 | **61.08** ±0.78 |
| $\mathcal{I}, \mathcal{G}, \mathcal{C}$ | Mean Time (s) (↓) | 13.80 | 23.28 | 14.55 | 14.64 | 18.68 | 14.53 |
| | GFLOPs (↓) | 59.05 | 59.59 | 59.06 | 59.32 | 59.74 | 59.06 |
| | # of Parameters (↓) | 34,424,162 | 40,185,923 | 36,504,643 | 37,960,643 | 340,929,475 | 36,685,511 |
| | Accuracy (↑) | 54.06 ±1.98 | 60.97 ±0.95 | 61.34 ±0.61 | 53.50 ±2.25 | 60.97 ±1.32 | **63.21** ±1.73 |
| $\mathcal{I}, \mathcal{G}, \mathcal{C}, \mathcal{B}$ | Mean Time (s) (↓) | 15.83 | 38.70 | 16.04 | 17.96 | 20.71 | 16.00 |
| | GFLOPs (↓) | 59.39 | 60.12 | 59.06 | 59.41 | 59.76 | 59.07 |
| | # of Parameters (↓) | 119,483,922 | 46,409,667 | 36,504,643 | 38,638,531 | 340,929,475 | 36,916,167 |
| | Accuracy (↑) | 57.24 ±0.35 | 58.82 ±0.82 | 61.44 ±1.16 | 55.18 ±2.42 | 59.52 ±1.00 | **66.11** ±1.14 |

Table 4: Complexity comparsion of mean time, GFLOPs, and # of parameters in ADNI dataset.

# 5 Conclusion

While multimodal learning brings new opportunities and challenges across various domains, including medical fields, existing approaches struggle to handle arbitrary modality combinations, especially in missing modality scenarios, often relying on single modalities or complete datasets. In this work, we propose a flexible multimodal learning framework, `Flex-MoE`, capable of managing arbitrary subsets of available modalities. By carefully considering modality combination, it leverages a learnable embedding bank to capture missing modality information and utilizes a unique SMoE design to enhance expert generalization and specialization. Extensive experiments on the representative ADNI and MIMIC-IV datasets validate its effectiveness in handling diverse modality combinations. Future work includes extending the framework to explore the scaling laws of available modalities, which in turn presents numerous modality combinations, offering significant room for further improvement.

**Societal Impact and Limitation:** The proposed algorithm has the potential to significantly improve early diagnosis and treatment outcomes for patients, reducing the burden on healthcare systems. However, its effectiveness can be limited by the availability of comprehensive and high-quality patient data, and there may be challenges in integrating this tool into existing clinical workflows.

**Acknowledgement** This work is supported by RF1-AG063481, R01-AG071174, Gemma Academic Program, and OpenAI Researcher Access Program.

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

# A Appendix

## A.1 Detailed Data Preprocessing in ADNI

| | File Name | Data Shape | Column Examples | Missing Rate (%) |
|---|---|---|---|---|
| Image | Processed T1-weighted fMRI | (10853, 91, 109, 91) | N/A | N/A |
| Genomics | ADNI_cluster_01_forward_757LONI | (757, 567386) | rs121434621', 'GSA-rs116587930' | 4.45 |
| | ADNI_GO_2_Forward_Bin | (432, 567386) | rs3131972', 'rs386134241' | 0.17 |
| | ADNI_GO2_GWAS_2nd_orig_BIN | (361, 567386) | rs182004761', 'rs386134241' | 0.29 |
| | ADNI3_PLINK_Final | (327, 567386) | rs3131972', 'rs3937033' | 0.62 |
| | ADNI3_PLINK_FINAL_2nd | (328, 567386) | 200610-37', 'rs386134241' | 0.46 |
| Clinical | MEDHIST_09May2024.csv | (3083,40) | MHSOURCE', 'MHPSYCH', 'MH2NEURL', 'MH3HEAD' | 13.28 |
| | NEUROEXM_09May2024.csv | (3873,28) | NXTREMOR', 'NXCONSCI','NXNERVE', 'NXMOTOR' | 13.71 |
| | PTDEMOG_09May2024.csv | (5073,77) | PTWORK', 'PTNOTRT','PTRTYR', 'PTHOME' | 66.58 |
| | RECCMEDS_09May2024.csv | (66622,29) | 'CMROUTE', 'CMREASON', 'CMEVNUM', 'CMBGN' | 35.46 |
| | VITALS_09May2024.csv | (15381,26) | VSHEIGHT', 'VSHTUNIT', 'VSBPSYS', 'VSBPDIA' | 20.98 |
| Biospecimen | APOERES_09May2024.csv | (2737, 17) | APTESTDT', 'APGEN1','APGEN2', 'APVOLUME' | 39.8 |
| | UPENNBIOMK_ROCHE_ELECSYS_09May2024.csv | (3174, 12) | ABETA40', 'ABETA42', 'TAU', 'PTAU' | 13.22 |

Table 5: Summary of data files with their respective shapes, column examples, and missing rates.

**Image Modality** We first performed magnetic field intensity inhomogeneity correction to ensure that the MRI images are uniform and reliable. Then, we used a method called MUSE (Multiatlas Region Segmentation Utilizing Ensembles of Registration Algorithms and Parameters) to segment the gray matter tissue, which was the focus of our study [14]. This process involves using multiple atlases and selecting the most optimal ones to accurately extract region-of-interest values from the segmented gray matter tissue maps. Next, voxelwise regional volumetric maps for each tissue volume were generated by spatially aligning skull-stripped images to a template residing in the Montreal Neurological Institute (MNI) space using a registration method [47].

**Genetic Modality** We collected SNP (single nucleotide polymorphisms) data from the ADNI 1, GO/2, and 3 studies. First, SNP data from the different phases of these studies were aligned to the same reference build using Liftover `https://liftover.broadinstitute.org/`. Specifically, all SNP data were converted to NCBI build 38 (UCSC hg38). After liftover, we merged the studies into a unified dataset. Next, linkage disequilibrium (LD) pruning, with parameters (50, 5, 0.1), was applied to filter out SNPs that were highly correlated with others. Here, the parameters represent the window size (50), the step size (5), and the r-squared threshold (0.1). The SNP data contains values of $0, 1, 2$.

**Biospecimen Modality** We extract biospecimen data from the following csv files provided by ADNI. The file UPENNBIOMK_ROCHE_ELECSYS_09May2024.csv was used for Total Tau and Phosphorylated Tau data. The file APOERES_09May2024.csv was used for ApoE genotype data. For numerical data, we applied a MinMax scaler to scale the values to a range of -1 to 1. For categorical data, we used one-hot encoding. For the missing values, we imputed the mean value for numerical columns and the mode for categorical columns.

**Clinical Modality** We extracted clinical data from the following csv files provided by ADNI: MEDHIST_09May2024.csv, NEUROEXM_09May2024.csv, PTDEMOG_09May2024.csv, REC-CMEDS_09May2024.csv, VITALS_09May2024.csv. During the preprocessing of this clinical data, we excluded the columns 'PTCOGBEG,' 'PTADDX,' and 'PTADBEG' as they contain information directly related to Alzheimer's Disease diagnosis. For numerical data, we applied a MinMax scaler to scale the values to a range of -1 to 1. For categorical data, we used one-hot encoding. For the missing values, we imputed the mean value for numerical columns and the mode for categorical columns.

## A.2 Hyperparameter Setting

Table 6: The hyperparameter setup for `Flex-MoE`.

| | ADNI | MIMIC-IV |
|---|---|---|
| | $\mathcal{I}, \mathcal{G}, \mathcal{C}, \mathcal{B}$ | $\mathcal{L}, \mathcal{N}, \mathcal{C}$ |
| Learning rate | 0.0001 | 0.0001 |
| # of Experts | 16 | 32 |
| # of SMoE layers | 1 | 1 |
| Top-K | 4 | 3 |
| Training Epochs | 50 | 50 |
| Warm-up Epochs | 5 | 5 |
| Hidden dimension | 128 | 128 |
| Batch Size | 8 | 8 |
| # of Attention Heads | 4 | 4 |

## A.3 More Primary Results

Table 7: Performance on ADNI dataset with Macro F1 metric across different models and modality combinations, given the Image ($\mathcal{I}$, 🧠), Genetic ($\mathcal{G}$, 🧬), Clinical ($\mathcal{C}$, ▦), and Biospecimen ($\mathcal{B}$, 🔬) modalities. $\mathcal{MC}$ denotes observed modality combination.

| $\mathcal{MC}$ | Modalities 🧠 | 🧬 | ▦ | 🔬 | [59] | [33] | ShaSpec | mmFormer | TF | MulT | MAG | LIMoE | FuseMoE | Flex-MoE |
|---|---|---|---|---|---|---|---|---|---|---|---|---|---|---|
| $\mathcal{I},\mathcal{G}$ | • | • | | | 52.92 ±0.17 | 52.75 ±3.91 | 45.86 ±1.55 | 40.25 ±7.03 | 60.03 ±0.88 | 60.59 ±1.05 | **61.06** ±0.79 | 58.75 ±0.83 | 61.04 ±0.95 | 61.05 ±1.03 |
| $\mathcal{I},\mathcal{C}$ | • | | • | | 25.65 ±6.58 | **57.36** ±1.45 | 47.68 ±0.80 | 44.31 ±8.97 | 54.45 ±0.90 | 51.88 ±1.30 | 52.72 ±3.39 | 51.84 ±0.96 | 53.32 ±1.47 | 54.16 ±2.01 |
| $\mathcal{I},\mathcal{B}$ | • | | | • | 28.78 ±1.32 | 57.93 ±2.04 | 49.82 ±2.00 | 45.82 ±9.33 | 51.20 ±2.64 | 52.64 ±2.57 | 53.36 ±2.96 | 52.70 ±3.47 | 50.38 ±1.31 | **58.50** ±0.94 |
| $\mathcal{G},\mathcal{C}$ | | • | • | | 50.54 ±1.38 | 51.62 ±3.34 | 50.29 ±0.36 | 39.45 ±5.47 | 27.85 ±1.89 | 36.77 ±6.42 | 29.33 ±0.46 | 29.57 ±3.65 | 27.49 ±2.87 | **59.44** ±0.49 |
| $\mathcal{G},\mathcal{B}$ | | • | | • | 31.21 ±1.71 | 51.41 ±4.30 | 56.32 ±4.83 | 35.63 ±1.44 | 29.82 ±1.48 | 32.41 ±1.55 | 28.29 ±1.05 | 20.91 ±0.60 | 20.91 ±0.60 | **61.65** ±1.71 |
| $\mathcal{C},\mathcal{B}$ | | | • | • | 24.78 ±6.24 | **63.46** ±0.35 | 55.46 ±4.06 | 33.09 ±5.43 | 29.57 ±1.99 | 33.22 ±0.72 | 27.20 ±3.53 | 20.36 ±0.00 | 29.11 ±3.83 | 59.13 ±1.75 |
| $\mathcal{I},\mathcal{G},\mathcal{C}$ | • | • | • | | 48.59 ±5.44 | 53.86 ±3.84 | 48.99 ±2.59 | 26.88 ±9.21 | 54.31 ±0.88 | 61.82 ±0.21 | 61.07 ±1.04 | 51.33 ±1.38 | 61.30 ±1.07 | **61.98** ±1.04 |
| $\mathcal{I},\mathcal{G},\mathcal{B}$ | • | • | | • | 55.46 ±2.05 | 54.82 ±4.18 | 51.80 ±0.99 | 28.19 ±8.29 | 53.70 ±2.43 | 52.46 ±1.64 | 52.54 ±1.64 | 52.80 ±1.92 | 52.75 ±0.91 | **59.45** ±3.14 |
| $\mathcal{I},\mathcal{C},\mathcal{B}$ | • | | • | • | 25.53 ±4.83 | 61.34 ±2.48 | 51.80 ±0.99 | 27.87 ±9.08 | 52.64 ±1.49 | 50.14 ±0.84 | 52.02 ±2.22 | 52.41 ±1.31 | 50.61 ±0.47 | **61.60** ±1.46 |
| $\mathcal{G},\mathcal{C},\mathcal{B}$ | | • | • | • | 24.95 ±6.49 | 60.57 ±2.64 | 60.57 ±2.64 | 25.99 ±9.98 | 40.40 ±6.88 | 29.38 ±0.79 | 31.59 ±1.93 | 27.99 ±2.13 | 27.49 ±0.93 | **64.15** ±1.69 |
| $\mathcal{I},\mathcal{G},\mathcal{C},\mathcal{B}$ | • | • | • | • | 49.76 ±1.95 | 57.93 ±2.04 | 51.80 ±0.99 | 53.64 ±9.09 | 57.27 ±0.44 | 59.58 ±0.77 | 61.38 ±1.32 | 53.63 ±0.30 | 59.55 ±1.60 | **64.73** ±2.01 |

Table 8: Performance on ADNI dataset with AUC metric across different models and modality combinations, given the Image ($\mathcal{I}$, 🧠), Genetic ($\mathcal{G}$, 🧬), Clinical ($\mathcal{C}$, ▦), and Biospecimen ($\mathcal{B}$, 🔬) modalities. $\mathcal{MC}$ denotes observed modality combination.

| $\mathcal{MC}$ | Modalities 🧠 | 🧬 | ▦ | 🔬 | [59] | [33] | ShaSpec | mmFormer | TF | MulT | MAG | LIMoE | FuseMoE | Flex-MoE |
|---|---|---|---|---|---|---|---|---|---|---|---|---|---|---|
| $\mathcal{I},\mathcal{G}$ | • | • | | | 70.04 ±0.72 | 70.25 ±3.26 | 66.07 ±1.11 | 68.05 ±2.04 | 73.45 ±1.06 | 70.95 ±1.76 | 73.14 ±0.71 | 71.88 ±1.14 | 72.37 ±1.08 | **74.52** ±1.81 |
| $\mathcal{I},\mathcal{C}$ | • | | • | | 54.52 ±2.93 | **73.99** ±1.02 | 65.39 ±0.82 | 68.15 ±2.09 | 72.88 ±0.81 | 71.37 ±1.65 | 71.68 ±1.90 | 71.86 ±1.27 | 70.98 ±0.24 | 73.03 ±0.14 |
| $\mathcal{I},\mathcal{B}$ | • | | | • | 57.02 ±8.74 | 76.06 ±0.85 | 68.86 ±0.69 | 68.44 ±1.98 | 71.70 ±0.42 | 72.43 ±1.84 | 72.82 ±1.84 | 71.82 ±1.65 | 70.59 ±1.09 | **77.68** ±0.33 |
| $\mathcal{G},\mathcal{C}$ | | • | • | | 69.24 ±0.45 | 66.46 ±3.29 | 71.38 ±0.82 | 65.50 ±5.45 | 47.57 ±1.96 | 61.17 ±5.61 | 52.00 ±1.08 | 51.14 ±0.60 | 49.23 ±1.54 | **78.34** ±0.47 |
| $\mathcal{G},\mathcal{B}$ | | • | | • | 48.91 ±5.97 | 69.68 ±3.58 | 75.29 ±2.91 | 64.69 ±5.26 | 51.32 ±2.33 | 53.53 ±0.68 | 51.36 ±1.07 | 51.82 ±0.30 | 51.82 ±0.30 | **79.24** ±0.79 |
| $\mathcal{C},\mathcal{B}$ | | | • | • | 58.41 ±5.16 | 79.53 ±0.34 | 78.73 ±0.22 | 63.25 ±5.89 | 49.82 ±2.03 | 64.36 ±2.80 | 50.20 ±1.63 | 48.29 ±3.21 | 48.82 ±0.58 | **79.65** ±0.81 |
| $\mathcal{I},\mathcal{G},\mathcal{C}$ | • | • | • | | 69.07 ±3.39 | 76.05 ±0.86 | 66.70 ±4.15 | 66.35 ±0.86 | 74.24 ±0.62 | 71.87 ±0.84 | 72.77 ±1.21 | 70.98 ±1.06 | 71.14 ±0.83 | **79.55** ±1.69 |
| $\mathcal{I},\mathcal{G},\mathcal{B}$ | • | • | | • | 70.75 ±2.30 | 76.02 ±0.86 | 69.79 ±1.39 | 65.91 ±2.01 | 72.11 ±2.08 | 71.88 ±0.34 | 72.35 ±0.32 | 71.70 ±0.81 | 72.16 ±0.57 | **79.27** ±0.65 |
| $\mathcal{I},\mathcal{C},\mathcal{B}$ | • | | • | • | 52.98 ±5.16 | 76.06 ±0.85 | 69.79 ±1.39 | 66.09 ±1.67 | 72.04 ±0.92 | 71.32 ±0.16 | 71.97 ±1.76 | 72.25 ±0.65 | 71.20 ±1.33 | **80.55** ±1.26 |
| $\mathcal{G},\mathcal{C},\mathcal{B}$ | | • | • | • | 56.71 ±4.72 | 70.06 ±0.85 | 79.18 ±0.63 | 63.49 ±4.78 | 69.00 ±0.66 | 60.85 ±5.25 | 51.73 ±0.41 | 49.36 ±0.52 | 48.82 ±1.02 | **81.67** ±0.59 |
| $\mathcal{I},\mathcal{G},\mathcal{C},\mathcal{B}$ | • | • | • | • | 69.44 ±0.84 | 76.06 ±0.85 | 69.79 ±1.39 | 73.93 ±5.97 | 69.42 ±3.20 | 71.09 ±0.66 | 71.99 ±0.54 | 72.05 ±0.27 | 71.16 ±1.01 | **81.67** ±0.54 |

Table 9: Performance on MIMIC-IV dataset with Macro F1 metric across different models and modality combinations, given the Lab and Vital values ($\mathcal{L}$, 💉), Clinical Notes ($\mathcal{N}$, 📄), and ICD-9 Codes ($\mathcal{C}$, 📋) modalities. $\mathcal{MC}$ denotes observed modality combination.

| $\mathcal{MC}$ | Modalities 💉 | 📄 | 📋 | TF | MulT | MAG | LIMoE | FuseMoE | Flex-MoE |
|---|---|---|---|---|---|---|---|---|---|
| $\mathcal{L},\mathcal{N}$ | • | • | | 50.81 ±0.47 | 50.61 ±2.77 | **53.46** ±0.26 | 55.19 ±1.52 | 52.79 ±1.32 | 51.29 ±1.83 |
| $\mathcal{L},\mathcal{C}$ | • | | • | 55.09 ±1.29 | **56.33** ±1.00 | 54.07 ±0.98 | 57.32 ±0.52 | 54.78 ±0.91 | 53.85 ±1.43 |
| $\mathcal{N},\mathcal{C}$ | | • | • | 54.37 ±0.41 | 55.33 ±1.04 | 54.15 ±1.79 | 54.59 ±0.65 | **55.54** ±0.60 | 53.02 ±3.99 |
| $\mathcal{L},\mathcal{N},\mathcal{C}$ | • | • | • | 54.19 ±0.38 | **58.43** ±0.22 | 55.04 ±1.41 | 55.79 ±0.94 | 55.38 ±1.06 | 53.19 ±1.28 |

Table 10: Performance on MIMIC-IV dataset with AUC metric across different models and modality combinations, given the Lab and Vital values ($\mathcal{L}$, 💉), Clinical Notes ($\mathcal{N}$, 📄), and ICD-9 Codes ($\mathcal{C}$, 📋) modalities. $\mathcal{MC}$ denotes observed modality combination.

| $\mathcal{MC}$ | Modalities 💉 | 📄 | 📋 | TF | MulT | MAG | LIMoE | FuseMoE | Flex-MoE |
|---|---|---|---|---|---|---|---|---|---|
| $\mathcal{L},\mathcal{N}$ | • | • | | 56.31 ±1.00 | 57.10 ±0.78 | 58.11 ±0.83 | 62.64 ±1.81 | 58.33 ±0.36 | **64.39** ±0.28 |
| $\mathcal{L},\mathcal{C}$ | • | | • | 60.43 ±0.76 | 65.12 ±2.19 | 60.75 ±0.20 | 65.14 ±0.34 | 62.52 ±0.39 | **66.27** ±0.17 |
| $\mathcal{N},\mathcal{C}$ | | • | • | 61.37 ±1.33 | 62.18 ±0.77 | 61.99 ±0.25 | 61.34 ±0.41 | 61.74 ±0.31 | **64.27** ±0.87 |
| $\mathcal{L},\mathcal{N},\mathcal{C}$ | • | • | • | 60.66 ±0.65 | 67.35 ±0.18 | 61.29 ±0.32 | 65.18 ±0.60 | 61.67 ±0.15 | **69.87** ±0.81 |

