# OpenReview forum: "Flex-MoE: Modeling Arbitrary Modality Combination via the Flexible Mixture-of-Experts"
_NeurIPS.cc/2024/Conference — NeurIPS 2024 spotlight_

### Official Review · Reviewer_6yyS · 2024-07-12

**Soundness:** 4
**Presentation:** 3
**Contribution:** 3
**Rating:** 7
**Confidence:** 4

**Summary:**

The paper aims to deal with the task of AD patient classification with missing modalities, and proposes a framework, Flex-MoE.
Flex-MoE also utilizes a learnable missing modality combination bank to complete the missing embeddings of missing modalities.
The training strategy and the gating mechanism of Flex-MoE are also well-designed to handle this task.
The experiments verify that Flex-MoE is able to achieve SoTA performance in this task on the shown modality combinations.

**Strengths:**

- The task explored by this study is practical in real-world scenarios. Addressing the issue of missing modalities in the AD domain is worth the attention of the community.
- The introduction section is well-written and easy to follow. (suggestion: using .pdf or .svg format for the figures is recommended.)
- The motivations for the proposed techniques in Flex-MoE, which you have claimed in your paper, are rational and decent.
- The ablation and sensitivity study is sufficient, showcasing the effectiveness of the components in Flex-MoE.

**Weaknesses:**

- The expression from Line 159 to Line 167 is a little bit confusing. More explanation (or even illustration) is needed.
- Is the usage of the missing modality bank just a simple look-up operation? I am concerned that it seems to be naive and there may be a more smart approach to achieve this.
- There is not $R_{(x_j)}$ of Line 199 in your equations. You need to correct this.
- Why the S-router can be trained to activate the corresponding expert index by Eq. (3)?
- It would be better if you could use some equations to assist to express the content of Line  204-215.
- This paper also leaks sufficient discussion of some related works, such as other tasks of biomedical images [1,2], in Sec.2 or appendix.
- The modality combinations of the experiments in Table 2 is few. It would be better to provide the experiments based on more combinations to showcase the performance of Flex-MoE.

[1] Zhou, D., Gu, C., Xu, J., Liu, F., Wang, Q., Chen, G., & Heng, P. A. (2023). RepMode: learning to re-parameterize diverse experts for subcellular structure prediction. In Proceedings of the IEEE/CVF Conference on Computer Vision and Pattern Recognition (pp. 3312-3322).

[2] Jiang, Y., & Shen, Y. (2024). M$^4$oE: A Foundation Model for Medical Multimodal Image Segmentation with Mixture of Experts. arXiv preprint arXiv:2405.09446.

**Questions:**

Please see the section of weaknesses.
I may raise or lower my score after the rebuttal.

**Limitations:**

The authors have claimed the limitations and broader impacts of the proposed method.

---

> ### Author Rebuttal · Authors · 2024-08-07
>
> We thank reviewer **6yyS** for mentioning our work as `practical`, `addressing missing modalities is worth attention to the AD community` and `rationale and decent techniques`. For the concerns, we provide responses below:
>
> ---
>
> **[W 1, 3: Expression from Line 159 - 167, 199]**
> The main idea of the missing modality bank completion is to supplement missing modalities from a predefined bank, ensuring robust data integration with observed ones. For example, if a patient lacks clinical data but has imaging, biospecimen, and genetic data, the observed modalities pass through their specific encoders. The missing clinical embedding is supplemented from the missing modality bank, indexed by the observed modalities (e.g., {Imaging, Biospecimen, Genetic}, Clinical). This approach prevents reliance on incomplete or naively imputed data, as encoders only process observed modalities. Once embeddings are standardized in dimension, they proceed to the Sparse MoE layer within the Transformer architecture. For line 199,  it should be $\text{max}(\mathcal{S}\text{-Router}(x_j))$. To improve readability, we will include such details in the final version.
>
> ---
>
> **[W 2: Usage of Missing Modality Bank]**
> As the reviewer mentioned, this can be seen as a look-up operation, but the way we construct and leverage this look-up table is what makes our work novel and distinctive. The rationale behind designing the missing modality bank is to differentiate the context of patient groups based on their unique observed modality combinations and corresponding missing modalities. In the AD domain, the missing modality problem provides a unique context for each patient’s diagnosis. For example, patients with biospecimen and image data but missing clinical and genetic data exhibit unique characteristics [1]. Motivated by this, we designed the missing modality bank with learnable embeddings indexed by observed modality combinations and their corresponding missing modality. This allows existing encoders to process only the observed features without being distracted by missing samples, equipping the model to handle various observed modality contexts flexibly. Empirically, as shown in Figure 4 of the manuscript, the clinical modality (C) shows more significant similarities with the full modality combination when predicting AD diagnosis compared to other modalities (I, G, B). Additionally, patients missing clinical (C) and genetic (G) modalities exhibit more similarities compared to other missing modality combinations, reinforcing our claim that missing modality combinations provide unique diagnostic contexts in AD.
>
> [1] https://pubmed.ncbi.nlm.nih.gov/24360540/
>
> ---
>
> **[W 4: Equation to assist line 204 - 215]**
> Here, we aim to utilize load balancing loss [1] especially targeted for the remaining experts (i.e., experts except top-1 selected expert) to ensure experts for each modality combination to be balanced activated. Formally it can be expressed as,
>
> 1. **Load Balancing Loss**
>    $\mathcal{L}_{\text{balance}} = \text{CV}^2(\sum{_j^N} \text{importance}{_j}) + \text{CV}^2(\sum{_j^N} \text{load}{_j}) $
> - where $N$ is the number of samples
>
> 2. **Coefficient of Variation Squared (CV²)**
>    $ \text{CV}^2(x) = \left( \frac{\sigma(x)}{\mu(x)} \right)^2 $
> -    where $\sigma(x)$ is the standard deviation of $x$ and $\mu(x)$ is the mean of $x$.
>
> 3. **Importance and Load**
>   $\text{importance}{_e} = \sum{_j^N} g _ {je},  \forall e \in E \setminus \{e _ {\text{top-1}}\}  $
>  - where $e$ is expert index and $g{_ie}$ is the gate value for sample $j$ with expert $e$
>  $\text{load}{_e} = \sum{_j^N} \delta(g _ {je} > 0),  \forall e \in E \setminus \{e _ {\text{top-1}}\}  $
>  - where $\delta(g _ {je} > 0)$ is an indicator function that is 1 if the gate value $g _ {ie}$ is greater than 0, indicating that the expert $e$ is selected for sample $j$.
>
> For the prediction head, we can simply denote it as: $\mathbf{Y} = \mathbf{Z} \cdot \mathbf{W}$ where
> $\mathbf{Y} \in \mathbb{R}^{N \times |\mathcal{C}|} $ denote the predicted probability for each sample on each class, $\mathbf{Z} \in \mathbb{R}^{N \times (D*|\mathcal{M}|)}$ denote the concatenated embedding of each modality, and $ \mathbf{W} \in\mathbb{R}^{(D*|\mathcal{M}|) \times |\mathcal{C}|}$ denote the weight matrix that transforms the concatenated embedding into final prediction space.
>
> For the final version, we will incorporate above equations to provide better understanding.
>
> [1] https://arxiv.org/abs/1701.06538
>
> ---
>
> **[W 5: Discussion of related works]**
> To briefly summarize, RepMode predicts 3D fluorescent images of subcellular structures from 3D transmitted-light images, addressing challenges of partial labeling and multi-scale variations. M4oE uses modality-specific experts and a gating mechanism to enhance medical image segmentation across various modalities. However, RepMode focuses on subcellular structure prediction, and M4oE focuses on medical image segmentation, differing from our work, which aims to accurately classify the stage of AD disease, particularly in the multimodal AD domain. While these works leverage image modalities, they lack the ability to cope with various other modalities, such as genetic and clinical. Moreover, although M4oE uses modality-specific experts, it does not consider the characteristic of modality combination information in practical missing modality scenarios, which is our main focus.
>
> ---
>
> **[W 6: More modality combination experiments]**
> To achieve greater generalizability, besides more modality combinations, we added the MIMIC dataset [1], the AUC score as an additional metric, and two more baselines (ShaSpec and mmFormer) during the rebuttal period. The results can be found here: [PDF](https://openreview.net/attachment?id=KkGeR9PVFe&name=pdf).
>
> We still observe Flex-MoE outperforms existing baselines across various modality combinations.
>
> [1] https://physionet.org/content/mimiciv/3.0/

---

> > ### Comment · Reviewer_6yyS · 2024-08-12
> > **The authors have addressed my concerns well**
> >
> > Dear authors and AC,
> >
> > Sorry for my late reply.
> > The authors have addressed my concerns, with additional experiments, explanations, and discussion for some unclear expression in the paper.
> > I would like to raise my score to 7, if the authors can add these contents to the main paper or the appendix.
> > Thanks for your time and effort.
> >
> > Best,
> > Reviewer 6yys

---

> > > ### Author Response · Authors · 2024-08-12
> > > **Thank you for your consideration**
> > >
> > > Dear Reviewer **6yyS**,
> > >
> > > We are pleased to hear that our rebuttal has addressed your concerns. We will ensure that those additional experiments are included in the final version.
> > >
> > > Best wishes,
> > > Authors

---

> > > > ### Comment · Reviewer_6yyS · 2024-08-13
> > > > **Response from Reviewer 6yys**
> > > >
> > > > Dear authors,
> > > >
> > > > Please also make sure that you will include all explanations and discussions for these concerns in your final paper or appendix, which can decently improve the quality and clearness of this work.
> > > >
> > > > Best,
> > > >
> > > > Reviewer 6yys

---

> > > > > ### Author Response · Authors · 2024-08-14
> > > > > **Thank you and we will ensure inclusion of discussions**
> > > > >
> > > > > Dear Reviewer **6yys**,
> > > > >
> > > > > Thank you for your response.
> > > > > As noted by the reviewer, we will ensure that the following points are included in our final version:
> > > > >
> > > > > - 1. **More detailed explanations** regarding lines 159-167 and the equations in lines 204-215.
> > > > > - 2. **Additional modality combination** results, expanding beyond those currently presented ($\mathcal{I}$, $\mathcal{I}+\mathcal{G}$, $\mathcal{I}+\mathcal{G}+\mathcal{C}$, $\mathcal{I}+\mathcal{G}+\mathcal{C}+\mathcal{B}$).
> > > > > - 3. Inclusion of **more baseline** discussions and results, specifically for ShaSpec and mmFormer.
> > > > > - 4. Incorporation of **additional datasets**, such as the MIMIC dataset.
> > > > > - 5. Analysis of **computational efficiency**, including training time, GFLOPs, and parameters.
> > > > >
> > > > > Once again, thank you for all your insightful comments to improve our paper.
> > > > >
> > > > >
> > > > > Best,
> > > > > Authors

---

> ### Comment · Area_Chair_agHq · 2024-08-08
> **Please read the rebuttal to check if the authors addressed your concerns**
>
> Dear Reviewer 6yyS,
>
> Can you have a look at the rebuttal and see if your concerns have been addressed?
>
> Best regards
> Your AC.

---

> > ### Author Response · Authors · 2024-08-10
> > **Eager for Your Feedback on Our Rebuttal**
> >
> > Dear Reviewer **6yyS**,
> >
> > We sincerely thank you for dedicating your time to review our work and for your constructive feedback. As the deadline for the discussion period approaches, we are eager to engage further and understand if our responses address your concerns satisfactorily.
> >
> > For quick access to additional experiments, including `more baseline results, datasets, and computational efficiency`, you can find them here: [PDF](https://openreview.net/attachment?id=KkGeR9PVFe&name=pdf)
> >
> > We would greatly appreciate it if you could kindly review our response. Thank you for your consideration.
> >
> > Best,
> > Authors

---

> ### Comment · Area_Chair_agHq · 2024-08-11
> **Please check the authors' rebuttal**
>
> Dear Reviewer 6yyS,
>
> Please don't forget to read the authors' rebuttal to reach a final decision about this paper in the next day or so. If you have any further questions to the authors, that would be a great chance to reach out to them.
>
> Best regards
> Your AC.

---

### Official Review · Reviewer_qdnn · 2024-07-12

**Soundness:** 2
**Presentation:** 2
**Contribution:** 2
**Rating:** 5
**Confidence:** 3

**Summary:**

The paper presents a multimodal learning framework, Flex-MoE (Flexible Mixture-of-Experts), designed to integrate diverse modalities in Alzheimer's Disease (AD) research using a Sparse Mixture-of-Experts design. Flex-MoE sorts samples based on the number of available modalities and processes them through modality-specific encoders. The framework trains experts with full modality samples and uses an S-Router to adapt the knowledge for fewer modality combinations.

**Strengths:**

1. The research topic addressing missing modalities in Alzheimer’s disease is crucial and highly relevant.
2. Source code is provided, facilitating the replication of experiments.

**Weaknesses:**

1. The model's specificity to Alzheimer's disease is unclear, as it appears to be a general approach for handling missing modalities.
2. The experimental setup lacks clarity. Details on how labels (e.g., dementia, CN, MCI) were decided, the prediction timeline, and the handling of patients at different stages are not provided. Additionally, the use of clinical data and whether it is time series data is not explained.
3. The evaluation is unconvincing. The imbalance rate of the datasets is not discussed. Only accuracy and F1 score are used for evaluation, but additional metrics like AUC could provide a more comprehensive understanding of the model’s performance. The process for selecting the threshold for classification to obtain the F1 score is also unclear.
4. The computational resources required for training and inference using Flex-MoE are not specified, especially given the need to train the model for various modality combinations.
5. No external dataset is included to validate the model's performance.

**Questions:**

1. Why is the model specifically designed for Alzheimer's disease, and how does it differ from a general model for missing modalities?
2. How were the labels (dementia, CN, MCI) decided? What is the prediction timeline, and were patients at different stages treated as separate samples? What time period of data was used, and was the clinical data time series data?
3. What is the imbalance rate of the datasets? Including metrics like AUC can provide a more complete understanding of the model’s performance. How was the threshold chosen for determining the classification boundary to obtain the F1 score?
4. What specific computational resources are required for training and inference using Flex-MoE?
5. How would the model perform on an additional external dataset?

**Limitations:**

1. The evaluation and practical application of the model are challenging to assess. The necessity of training different models for each modality combination is not well justified.
2. The lack of external dataset validation limits the generalizability of the model's performance.

---

> ### Author Rebuttal · Authors · 2024-08-07
>
> We thank reviewer **qdnn** for the comprehensive review of our paper. Specifically, we appreciate the reviewer's recognition of our key contribution in addressing missing modalities in AD as `crucial` and `highly relevant`. For the concerns raised, we provide the following responses:
>
> ---
>
> **[W 1 & Q 1: model’s specificity to AD]**
> As the reviewer mentioned, our approach can be generalized, similar to common phenomena in various biomedical papers (e.g., Transformer in 3D MRI [1], Graph Neural Network in the single-cell domain [2,3]). However, we emphasize the importance of a careful application that considers domain-specific characteristics, especially in response to recent improvements in the ML field. In this paper, we focused on the AD domain for two main reasons:
>
> - **Underexplored Characteristic**: Despite the comprehensive multi-modal nature of AD (e.g., clinical, biospecimen, image, genetic) [3], it remains underexplored compared to domains with fewer modalities (e.g., image, text) [4, 5]. Empirically, as shown in Table 2 of the manuscript, the direct application of multimodal ML, e.g., LIMoE [5], underperforms compared to Flex-MoE in handling 3 or more modalities and in overall performance. This supports our motivation to address the diverse multimodal nature prevalent in the AD domain and the specific missing modality combinations.
>
> - **Unique Characteristics of Missing Modalities**: In the AD domain, the missing modality problem provides a unique context for each patient’s diagnosis. For instance, patients with biospecimen and image modalities but missing clinical and genetic data present unique characteristics [6]. Empirically, as shown in Figure 4 of the manuscript, clinical modality (C) shares more significant similarities with the full modality combination when predicting AD diagnosis compared to other modalities (I, G, B). Additionally, patients missing clinical (C) and genetic (G) modalities show more similarities compared to other missing modality combinations, reinforcing our claim that missing modality combinations provide unique diagnostic contexts in AD.
>
> In summary, our study aims to advance clinical diagnosis in the AD domain by effectively integrating multimodal data, especially in the growing era of AI4Science (Biology, Medicine, Health).
>
> [1] https://www.nature.com/articles/s41598-024-59578-3
> [2] https://www.nature.com/articles/s41467-021-22197-x
> [3] https://www.nature.com/articles/s41467-023-36559-0
> [4] https://arxiv.org/abs/2309.15857
> [5] https://arxiv.org/abs/2206.02770
> [6] https://pubmed.ncbi.nlm.nih.gov/24360540/
>
> ---
>
> **[W 2, 3 & Q 2: Clarity in Experimental Setup]**
> To validate the effectiveness of Flex-MoE, we focused on the gold-standard ADNI dataset. Following existing studies [1,2], we performed an AD stage prediction task to predict specific labels for Dementia, CN, and MCI. Specifically, we used the diagnostic summary file 'DXSUM_PDXCONV_22Apr2024.csv' and the 'DIAGNOSIS' column, which underwent clinical and neuropsychological evaluation as reported by ADNI [3]. The label statistics in our study were: CN (1030 patients, 43.3%), MCI (860 patients, 36.1%), Dementia (490 patients, 20.6%). Despite concerns about imbalance, we observed a relatively balanced ratio of 490/1030 = 0.47. The F1-score was calculated using the scikit-learn package [4], which automatically computes the score without manual selection of threshold.
>
> Regarding timeline information, timestamp data is not readily available across genetic, biospecimen, and clinical modalities. However, for the image modality, we selected the most recent image to ensure relevance and mapped the patient ID with other modalities. Thus, our method remains static, but dynamic analysis with time information is a promising direction for future work.
>
> We will add this information to the final version for clarity. Detailed statistics and preprocessing steps for each modality are provided in Section 4.1 and Appendix A of the manuscript, addressing the reviewer's concerns about the clinical modality.
>
> [1] https://www.nature.com/articles/s41598-020-74399-w
> [2] https://www.nature.com/articles/s41598-018-37769-z
> [3] https://adni.loni.usc.edu/wp-content/uploads/2008/07/inst_about_data.pdf
> [4] https://scikit-learn.org/stable/modules/generated/sklearn.metrics.f1_score.html
>
> ---
>
> **[W 4 & Q 4: Computational Resources]**
> We appreciate the reviewer's concerns about computational resources. We used NVIDIA A100 GPUs for training and inference but highlight the efficiency of adopting the Sparse MoE design, where only the selected top-k experts are utilized. We performed a computational comparison experiment in terms of Mean Time, FLOPs, and the number of activated parameters, which can be found here: [PDF](https://openreview.net/attachment?id=KkGeR9PVFe&name=pdf).
> The result demonstrate the efficiency of the SMoE design, showcasing the potential for various real-world datasets.
>
> ---
>
> **[W 3, 5 & Q 4, 5 & L 1, 2: More Comprehensive Experiment]**
> To achieve greater generalizability, we added the MIMIC dataset [1], the AUC as an additional metric, and two more baselines (ShaSpec and mmFormer) during the rebuttal period. We tested all possible modality combinations, and the results can be found here: [PDF](https://openreview.net/attachment?id=KkGeR9PVFe&name=pdf).
> Our experiments show that Flex-MoE outperforms existing baselines across various metrics and modality combinations, thanks to its careful consideration of modality combinations. Additionally, to justify the necessity of training different experts, Figure 5 shows that each expert index contains both global context knowledge and specialized modality combination knowledge, providing flexibility in responding to various modality combinations. Figure 4 further illustrates the interpretability of how different observed and missing modality combinations share similarities in predicting AD status.
>
> [1] https://physionet.org/content/mimiciv/3.0/

---

> ### Comment · Area_Chair_agHq · 2024-08-08
> **Please read the rebuttal to check if the authors addressed your concerns**
>
> Dear Reviewer qdnn,
>
> Can you have a look at the rebuttal and see if your concerns have been addressed?
>
> Best regards
> Your AC.

---

> ### Author Response · Authors · 2024-08-10
> **Eager for Your Feedback on Our Rebuttal**
>
> Dear Reviewer **qdnn**,
>
> We sincerely thank you for dedicating your time to review our work and for your constructive feedback. As the deadline for the discussion period approaches, we are eager to engage further and understand if our responses address your concerns satisfactorily.
>
> For quick access to additional experiments, including `more baseline results, datasets, and computational efficiency`, you can find them here: [PDF](https://openreview.net/attachment?id=KkGeR9PVFe&name=pdf)
>
> We would greatly appreciate it if you could kindly review our response. Thank you for your consideration.
>
> Best,
> Authors

---

> ### Comment · Area_Chair_agHq · 2024-08-11
> **Please check the authors' rebuttal**
>
> Dear Reviewer qdnn,
>
> Please don't forget to read the authors' rebuttal to reach a final decision about this paper in the next day or so. If you have any further questions to the authors, that would be a great chance to reach out to them.
>
> Best regards Your AC.

---

> > ### Comment · Reviewer_qdnn · 2024-08-12
> >
> > Thank you for the authors' response. I still have some concerns about the specific cognitive assessments included in the model. I’m willing to adjust my scores to 5 and my confidence to 3.

---

> > > ### Author Response · Authors · 2024-08-14
> > > **Appreciation and further clarification on cognitive assessment in ADNI dataset**
> > >
> > > Dear Reviewer **qdnn**,
> > >
> > > Thank you for your response. We appreciate that our rebuttal was well received, and we thank you for your willingness to increase the score.
> > >
> > > In our above rebuttal response [link](https://openreview.net/forum?id=ihEHCbqZEx&noteId=nYk2KzEUPG) under **[W 2, 3 & Q 2: Clarity in Experimental Setup]**, we provided the specific file referenced in this study, where the labels (e.g., Dementia, CN, MCI) for each patient were annotated according to the ADNI dataset. We also presented the statistical distribution and imbalance ratio.
> > >
> > > However, we realize that there may still be some concerns regarding the details of the **cognitive assessments** used in our model. These assessments are crucial in determining the diagnostic categories such as Cognitively Normal (CN), Mild Cognitive Impairment (MCI), and Dementia. Below, based on ADNI documentation [1], we provide a more detailed explanation of each label including the specific cognitive tests involved and how they contribute to the classification process.
> > >
> > > 1. **Cognitively Normal (CN) (Interchangeable with Cognitively Unimpaired (CU))**:
> > >     - **Meaning**: Participants in this group show no signs of significant cognitive decline at the time of their evaluation.
> > >    - **Key Cognitive Assessments**: To be classified as CN/CU, individuals must score 0 on the Clinical Dementia Rating (CDR) and show no memory impairment (memory box score of 0). Their performance on the Mini-Mental State Exam (MMSE) and the Wechsler Logical Memory II sub-scale must be within the normal range, adjusted for education level.
> > >    - **Significance**: These tests ensure that participants classified as CN/CU exhibit cognitive functioning consistent with normal aging, without evidence of dementia or mild cognitive impairment.
> > >
> > > 2. **Mild Cognitive Impairment (MCI)**:
> > >     - **Meaning**: MCI is an intermediate stage between normal aging and dementia, where participants show noticeable cognitive impairments that do not yet meet the criteria for dementia.
> > >    - **Key Cognitive Assessments**: Individuals are labeled as MCI if they have a CDR global score of 0.5 and a memory box score of at least 0.5. The MMSE and Wechsler Logical Memory II sub-scale are also used, with specific cutoffs to diagnose MCI based on educational levels. These cognitive tests reveal impairments in memory and other cognitive domains that are more pronounced than in normal aging but not severe enough to warrant a dementia diagnosis.
> > >    - **Significance**: These assessments are crucial for identifying individuals at risk for developing dementia, making them a key focus for early interventions.
> > >
> > > 3. **Dementia (Dementia/AD)**:
> > >    - **Meaning**: Participants in this group meet the clinical criteria for dementia, typically due to Alzheimer's Disease, characterized by significant cognitive and functional impairment.
> > >    - **Key Cognitive Assessments**: A CDR global score of 0.5 or 1, combined with impaired scores on the MMSE and Wechsler Logical Memory II sub-scale, typically categorizes a participant as having dementia. Additionally, the classification process involves screening to exclude those whose symptoms suggest non-Alzheimer's forms of dementia, such as Frontotemporal Dementia.
> > >     - **Significance**: These cognitive assessments are vital for studying the progression of Alzheimer's Disease and for developing treatments targeted at this advanced stage.
> > >
> > > For further details regarding the protocols and ADNI study information (we used ADNI 3 in this study), you may kindly refer to the official ADNI documentation [2].
> > >
> > > In summary, to provide a clearer understanding for readers, we will incorporate this discussion into our final version.
> > >
> > > Once again, thank you for all your comments to improve our paper.
> > >
> > > Best,
> > > Authors
> > >
> > > [1] https://adni.loni.usc.edu/data-samples/adni-data/study-cohort-information/
> > > [2] https://adni.loni.usc.edu/help-faqs/adni-documentation/

---

### Official Review · Reviewer_BwQM · 2024-07-24

**Soundness:** 3
**Presentation:** 4
**Contribution:** 3
**Rating:** 7
**Confidence:** 4

**Summary:**

This paper introduces Flex-MoE, a novel multimodal learning framework for Alzheimer's Disease that handles missing modalities using a Sparse Mixture-of-Experts design and demonstrates its efficacy on the ADNI dataset.

**Strengths:**

1. The idea of Flex-MoE is clear and straightforward, effectively addressing the challenge of integrating and handling missing modalities in Alzheimer's Disease research.
2. The expression is smooth and clear, making the complex concepts and methodologies easily understandable.

**Weaknesses:**

1. The paper emphasizes that the Sparse Mixture-of-Experts (SMoE) selectively activates only the most relevant experts to improve scalability, but does not provide experimental results demonstrating computational efficiency or scalability.

2. The interpretation of \( B_{M \setminus m, m} \) in Equation (2) is unclear; it should be better explained in details. Whether it involves a learnable module that outputs missing modality features based on available modality features, or via other method?

3. The approach to handling missing modalities differs from the method in the paper "Multi-modal Learning with Missing Modality via Shared-Specific Feature Modelling," which uses averaging for missing modalities. A comparison with this approach would be beneficial. The paper provides code https://github.com/billhhh/ShaSpec.

4. The paper overlooks some existing missing modality baselines, such as mmFormer and ShaSpec.

Some minor weaknesses:
1. The abstract states that "few recent studies attempt to integrate multiple modalities," but there are actually many methods for handling missing modalities (as noted in point 4 above), which are not compared experimentally in this paper.

2. The font size in Table 1 is too small and needs to be redrawn for better readability.

3. In line 180, the bold formatting for "It" is incorrect.

4. In Table 2(b), the value 61.08 in the first row is not the best result, which should be corrected.

5. Table 2 should ideally use a visualization approach similar to that in the ShaSpec paper to represent different modality combinations more clearly.

**Questions:**

The most biggest question for me is pointed in weaknesses Q2: The interpretation of \( B_{M \setminus m, m} \) in Equation (2) is unclear; it should be better explained in details. Whether it involves a learnable module that outputs missing modality features based on available modality features, or via other method?

**Limitations:**

Please refer to the weaknesses section

---

> ### Author Rebuttal · Authors · 2024-08-07
>
> We thank reviewer **BwQM** for their careful review and for mentioning that Flex-MoE is `straightforward`, `effectively addresses missing modalities in AD`, and `easily understandable`. For the remaining concerns, we provide details below:
>
> ---
>
> **[W 1: Computational efficiency and scalability]**
> Throughout the paper, we explained the rationale behind the SMoE design principle for handling missing modality scenarios by assigning modality combinations to each expert index. As the reviewer mentioned, utilizing the SMoE design indeed benefits computational efficiency and scalability. To further verify its effectiveness, we compared the training time, GFLOPs, and number of parameters used in training at this link: [PDF](https://openreview.net/attachment?id=KkGeR9PVFe&name=pdf).
>
> We observe that Flex-MoE not only performs best in the I, G, C, B modality configuration, as shown in Table 2 of the manuscript, but also achieves efficiency in terms of training time and GFLOPs. Additionally, it demonstrates scalability with fewer parameters compared to existing baselines, including the state-of-the-art model, FuseMoE [1].
>
> [1] https://arxiv.org/abs/2402.03226
>
> ---
>
> **[W 2 & Q 1: Interpretation of $\mathbf{B} _ {\mathcal{M} \setminus m}$ in Equation (2)]**
> The main idea of the missing modality bank completion is to supplement missing modalities from a learnable embedding bank. If a sample contains missing modality data, we retrieve the missing information (i.e., embedding) from this bank, which includes all possible modality combinations and their respective missing modalities. Otherwise, for the observed input, the data directly passes through the modality-specific encoder.
>
> For example, if a patient lacks clinical data (i.e., $m = \mathcal{C}$ where $m$ denotes the missing modality, thus $\mathcal{M} \setminus m = \{ \mathcal{I}, \mathcal{B}, \mathcal{G} \}$) but has imaging, biospecimen, and genetic data, the observed modalities pass through their respective encoders. The missing clinical embedding is supplemented from the missing modality bank, indexed by the observed modalities (i.e., $\mathbf{B} _ {\mathcal{M} \setminus m} = \mathbf{B} _ {\{\mathcal{I}, \mathcal{B}, \mathcal{G}\}}$). This approach ensures that encoders only process observed features, preventing them from being influenced by incomplete or naively imputed data. As this missing modality bank is learnable throughout the iterations, it eventually possesses the knowledge relevant to the downstream task, such as AD stage prediction.
>
> Empirically, as shown in Figure 4 of the manuscript, the clinical modality (C) shows more significant similarities with the full modality combination when predicting AD diagnosis compared to other modalities (I, G, B). Additionally, patients missing clinical (C) and genetic (G) modalities exhibit more similarities compared to other missing modality combinations, reinforcing our claim that missing modality combinations provide unique diagnostic contexts in AD.
>
> ---
>
> **[W 3 & W 4 & M-W 5: Comparison with other baselines in ShaSpec style]**
> We appreciate highlighting these two relevant baselines in our study. To briefly summarize each paper: ShaSpec employs a strategy using auxiliary tasks based on distribution alignment and domain classification, along with a residual feature fusion procedure, to learn shared and specific features for handling the missing modality issue. During the rebuttal period, we employed ShaSpec, and the performance comparison on the ADNI dataset with additional metric, AUC can be found here: [PDF](https://openreview.net/attachment?id=KkGeR9PVFe&name=pdf).
> Interestingly, while ShaSpec has a specific module to handle missing modalities, it falls short of Flex-MoE. We believe this is due to forcing distribution alignment without carefully considering the context of each modality combination. For instance, when the clinical modality is missing, ShaSpec leverages other modalities to generate the clinical feature, but this may overlook the unique contextual meaning of the missing clinical data. In contrast, Flex-MoE retrieves a learnable embedding from the missing modality bank to supplement the clinical embedding without forcing observed modalities to contribute, allowing the clinical embedding more flexibility to learn downstream task-specific information.
>
> Moreover, mmFormer is an end-to-end framework initially designed for different MRI modalities, consisting of hybrid modality-specific encoders, a modality-correlated encoder, and a convolutional decoder. This approach differs from ours, which considers images as one modality and extends the perspective to genetic, biospecimen, and clinical data. When implementing mmFormer for our task, we followed its proposed architecture for processing image data and replaced their encoder with our modality-specific encoders used in Flex-MoE. The results can be found here: [PDF](https://openreview.net/attachment?id=KkGeR9PVFe&name=pdf).
>
> In summary, mmFormer lacks the ability to handle diverse modalities beyond images, which have different contexts, making it significantly less effective than Flex-MoE. Additionally, by supplementing missing modality embeddings as global modality-invariant features, mmFormer fails to handle the specific context of modality combinations and corresponding missing modalities. This justifies the necessity of incorporating modality combination information while handling missing modalities, as proposed in Flex-MoE.
>
> ---
>
> **[M-W 1,2,3: Improvements]**
> Thank you for the constructive feedback. We will incorporate the discussed baselines, change the font size in Table 1 for better readability, and carefully revise the bold font.
>
> ---
>
> **[M-W 4: Best result in Table 2 (b)]**
> We have checked the bold formatting in Table 2 (b) and found no errors. Flex-MoE with 61.08 performs best in that row. Could we kindly ask reviewer **BwQM** to re-clarify the point? We are ready to provide more details and refine accordingly.

---

> > ### Comment · Reviewer_BwQM · 2024-08-14
> > **Adjusted the score to 7**
> >
> > I'd like to adjust my score to 7 according to the efforts of rebuttal by authors

---

> ### Comment · Area_Chair_agHq · 2024-08-08
> **Please read the rebuttal to check if the authors addressed your concerns**
>
> Dear Reviewer BwQM,
>
> Can you have a look at the rebuttal and see if your concerns have been addressed?
>
> Best regards
> Your AC.

---

> ### Author Response · Authors · 2024-08-10
> **Eager for Your Feedback on Our Rebuttal**
>
> Dear Reviewer **BwQM**,
>
> We sincerely thank you for dedicating your time to review our work and for your constructive feedback. As the deadline for the discussion period approaches, we are eager to engage further and understand if our responses address your concerns satisfactorily.
>
> For quick access to additional experiments, including `more baseline results, datasets, and computational efficiency`, you can find them here: [PDF](https://openreview.net/attachment?id=KkGeR9PVFe&name=pdf)
>
> We would greatly appreciate it if you could kindly review our response. Thank you for your consideration.
>
> Best,
> Authors

---

> ### Comment · Area_Chair_agHq · 2024-08-11
> **Please check the authors' rebuttal**
>
> Dear Reviewer BwQM,
>
> Please don't forget to read the authors' rebuttal to reach a final decision about this paper in the next day or so. If you have any further questions to the authors, that would be a great chance to reach out to them.
>
> Best regards Your AC.

---

> ### Comment · Reviewer_BwQM · 2024-08-12
> **Responses after rebuttal**
>
> The rebuttal of authors addresses all of my questions. They provide more baseline results, datasets, and computational efficiency. So I have no more questions related to the paper. W.r.t. the Best result in Table 2 (b), thank you for the clarification and I've double checked it again. It should be correct. Kindly add the new results in the newer version of the paper. Thanks for the effort!
>
> Best

---

> ### Author Response · Authors · 2024-08-12
> **Appreciation and kind request for your consideration of score adjustment**
>
> Dear Reviewer **BwQM**,
>
> Thank you once again for your thoughtful feedback and for confirming that all of your concerns have been addressed. We greatly appreciate your time and effort in reviewing our work.
>
> As you noted, we will include all modifications and results to our final version. We would be deeply grateful if you could kindly consider adjusting the score to reflect an improved positive evaluation of our submission. Your support at this stage would be of great help to us.
>
> Thank you once again for all your support.
>
> Best,
> Authors

---

### Author Rebuttal · Authors · 2024-08-07

In this study, we propose **Flex-MoE**, a novel framework designed to address the issue of missing modalities in the AD domain, where existing studies often **(1)** rely on single modality and complete data, and **(2)** overlook modality combinations. As a remedy, Flex-MoE includes a **missing modality bank completion** step, followed by **expert generalization and specialization** steps, and is equipped with a novel router design.

We express our profound gratitude to all reviewers for their time and effort in evaluating our manuscript. We particularly appreciate the positive feedback on the `clear and straightforward idea`, `effectively addressing missing modalities in AD domain which is crucial and highly relevant`, `worth the attention in the community`, `motivations for Flex-MoE are rational and decent`. Besides, all the constructive feedback has been invaluable in enhancing our work. There were common concerns raised by some reviewers, which we address below, alongside more detailed responses to each reviewer.

---

1. More comprehensive experiments - Reviewers **BwQM**, **qdnn**, **6yyS**
- To achieve greater generalizability, we added the (1) MIMIC dataset [1], the (2) AUC score as an additional metric, and (3) two more baselines (ShaSpec and mmFormer) during the rebuttal period. We tested (4) all possible modality combinations and the comprehensive result are shown in below [PDF](https://openreview.net/attachment?id=KkGeR9PVFe&name=pdf).
- The results show that in most cases, Flex-MoE consistently outperforms existing baselines across various metrics and modality combinations, thanks to its careful consideration of modality combinations. Notably, as the number of available modalities increases (e.g., 2, 3, 4), the performance of Flex-MoE significantly improves. In contrast, other baselines achieve their best performance when utilizing a subset of modalities. This demonstrates Flex-MoE's potential to effectively advance and address the challenges in the multi-modal AD domain.
- **Details on MIMIC dataset preprocessing**: For the MIMIC dataset, we use the Medical Information Mart for Intensive Care IV (MIMIC-IV) database, which contains de-identified health data for patients who were admitted to either the emergency department or stayed in critical care units of the Beth Israel Deaconess Medical Center in Boston, Massachusetts24. MIMIC-IV excludes patients under 18 years of age. We take a subset of the MIMIC-IV data, where each patient has at least more than 1 visit in the dataset as this subset corresponds to patients who likely have more serious health conditions. For each datapoint, we extract ICD-9 codes, clinical text, and labs and vital values. Using this data, we perform binary classification on one-year mortality, which foresees whether or not this patient will pass away in a year. We drop visits that occur at the same time as the patient's death. In order to align the experimental setup with the ADNI data, which does not contain temporal data, we take the last visit for each patient.

---

2. Computational Resources - Reviewers **BwQM**, **qdnn**
- Regarding the usage of SMoE design, we further performed a computational comparison experiment in terms of Mean Time, FLOPs, and the number of activated parameters, where the result can be found in below [PDF](https://openreview.net/attachment?id=KkGeR9PVFe&name=pdf).
- The results demonstrate the efficiency of the Sparse MoE design, showcasing the potential for improved performance on various real-world datasets.

---

3. Interpretation of Equation (2) - Reviewers **BwQM**, **6yyS**
- The main idea of the missing modality bank completion is to supplement missing modalities from a predefined bank, ensuring robust data integration with observed ones. For example, if a patient lacks clinical data but has imaging, biospecimen, and genetic data, the observed modalities pass through their specific encoders. The missing clinical embedding is supplemented from the missing modality bank, indexed by the observed modalities (e.g., {Imaging, Biospecimen, Genetic}, Clinical). This approach prevents reliance on incomplete or naively imputed data, as encoders only process observed modalities.

---

If you have any remaining concerns, we are more than happy to discuss them until the end of the discussion period. Please do not hesitate to ask any questions. Once again, we deeply appreciate all the careful reviews and the time from the reviewers.

Best,
Authors

---

### Decision · Program_Chairs · 2024-09-25

**Decision:**

Accept (spotlight)

**Comment:**

This paper addresses multimodal learning with missing modalities with Flex-MoE, which is a sparse mixture of experts method.  The approach is tested for Alzheimer’s disease classification using the ADNI dataset, and results show SOTA performance.  The reviewers identified several strengths, such as: 1) idea is clear, 2) paper is well written, 3) research topic is relevant, 4) motivation is well explained, and 5) the ablation study is comprehensive. The reviews also identified a few issues, which include: 1) missing more comprehensive experiments in other tasks, 2) missing discussion on complexity, 3) equation (2) needs clarification, and 4) missing comparison with other methods (e.g., mmformer, shaspec). The rebuttal addressed all points above and the final scores of the paper are 5,7,7 (average score of 6.33, with an average confidence of 3.67). I recommend this paper to be accepted.